# Convergent evolution of oxidized sugar metabolism in commensal and pathogenic microbes in the inflamed gut

Sophia Levy [1], Angela K. Jiang [2,3], Maggie R. Grant [1], Gabriela Arp [1], Glory Minabou Ndjite[1], Xiaofang Jiang [3] ✉ & Brantley Hall [1,2] ✉

Inflammation-associated perturbations of the gut microbiome are well characterized, but poorly understood. Here, we demonstrate that disparate taxa recapitulate the metabolism of the oxidized sugars glucarate and galactarate, utilizing enzymatically divergent, yet functionally equivalent, *gud/gar* pathways. The divergent pathway in commensals includes a putative 5-KDG aldolase (GudL) and an uncharacterized ABC transporter (GarABC) that recapitulate the function of their non-homologous counterparts in pathogens. A systematic bioinformatic search for the *gud/gar* pathway in gut microbes identified 887 species putatively capable of metabolizing oxidized sugars. Previous studies showed that inflammation-derived nitrate, formed by nitric oxide reacting with superoxide, promotes pathogen growth. Our findings reveal a parallel phenomenon: oxidized sugars, also produced from reactions with nitric oxide, serve as alternative carbon sources for commensal microbes. Previously considered a pathogen virulence factor, oxidized sugar metabolism is also present in specific commensals and may contribute to their increased relative abundance in gastrointestinal inflammation.

Inflammation-driven perturbations of the gut microbiome play an important role in the positive feedback circuit underlying gastro-intestinal inflammation, as exemplified in Inflammatory Bowel Disease (IBD), where microbial perturbations trigger an aberrant immunological response leading to the characteristic chronic inflammation and gastrointestinal symptoms[1,2]. This maladaptive interaction highlights the complex relationship between the commensal gut microbiome and the host immune system in the etiology of gut inflammation in both IBD and colitis[3,4].

The taxonomic changes associated with inflammation are well documented, but the mechanisms underlying these changes remain understudied. Shifts in taxonomic composition often include increases in the relative abundance of *Escherichia, Veillonella, Klebsiella, Blautia*, and *Lachnoclostridium* spp[5–8].

Recent studies have begun elucidating pathways responsible for these inflammation-induced alterations to the microbial community, including harmful effects of the oxidizing environment, increased abundance of terminal electron acceptors, and reduction of butyrate available to colonocytes[9–14]. Several key pathways trace back to the overexpression of *NOS2*, which encodes inducible nitric oxide synthase (iNOS), in the inflamed gut. *NOS2* expression is induced by transcription factor (NF)-kB during the inflammatory response[15]. Consequently, *NOS2* is highly upregulated during IBD. The Integrative Human Microbiome Project reported 4–12 fold higher *NOS2* expression in those with active flares of IBD compared to healthy individuals[16–20]. In colonic epithelial cells, *NOS2* expression produces nitric oxide radicals that diffuse directly into the lumen and spontaneously oxidize glucose and galactose to glucarate (saccharic acid)

[1]Department of Cell Biology and Molecular Genetics, University of Maryland, College Park, College Park, MD, USA. [2]Center for Bioinformatics and Computational Biology, University of Maryland, College Park, College Park, MD, USA. [3]National Library of Medicine, National Institutes of Health, Bethesda, MD, USA. ✉e-mail: xiaofang.jiang@nih.gov; brantley@umd.edu

and galactarate (mucic acid), respectively. PRISM metabolomics revealed a concordantly high abundance of oxidized sugars in the inflamed IBD gut[21,22].

Metabolism of oxidized sugars during inflammation has been implicated as a pathogen colonization factor. Faber et al. demonstrated enhanced colonization of *Salmonella enterica* in the inflamed murine gut via oxidized sugar metabolism, while genetic ablation of the oxidized sugar pathway reduced *S. enterica* competition[21].

The oxidized sugar metabolic pathway - hereafter referred to as the *gud/gar* pathway - enables bacteria to use glucarate and/or galactarate as a carbon source by catabolizing the oxidized sugars to pyruvate and 2-phosphoglycerate (2-PG), which can then be further metabolized through the remaining steps of canonical glycolysis[23,24]. During the penultimate stage of glycolysis, enolase converts 2-PG to phosphoenolpyruvate, which pyruvate kinase subsequently turns into pyruvate, producing ATP[25,26].

The well-characterized *E. coli gud/gar* pathway contains five core genes: glucarate/galactarate permeases (*gudP*/*garP*) import the oxidized sugars, glucarate/galactarate dehydratases (*gudD*/*garD*) convert the oxidized sugars into a common 5-keto-4-deoxy-D-glucarate (5-KDG) intermediate, 5-KDG aldolase (*garL*) splits 5-KDG into pyruvate and tartronate semialdehyde (TSA), TSA dehydrogenase (*garR*) forms glycerate, and, as a final step, glycerate kinase (*garK*) produces 2-PG (Fig. 1A)[23,27,28]. Despite the thorough characterization of the *gud/gar* pathway in *Enterobacteriaceae* spp., it is unknown whether other inflammation-adapted microbes can also utilize this pathway.

Here, we sought to systematically investigate the breadth of the diversity of commensal gut microbes that metabolize oxidized sugars. We found that several Bacillota (previously Firmicutes) species with increased relative abundance during IBD were able to use oxidized sugars as a carbon source. We searched for the *gud/gar* pathway in these species and found that while some genes homologous to those in the Enterbacteriaceae pathway - *gud/garD* and *garR* - were present, other core pathway genes like the permease *gud/garP* and aldolase *garL* were conspicuously missing. However, we hypothesized that unannotated genes within the same operons as the homologous genes might underlie the ability of these species to metabolize oxidized sugars. We individually tested the function of each gene in single-knockout *E. coli*, and observed that genes from *Enterocloster clostridioformis* (previously known as *Clostridium clostridioforme*), including an uncharacterized ABC transporter and aldolase, were able to rescue the corresponding knockout. Though *E. clostridioformis* contains no *garL* homolog, the uncharacterized aldolase was able to restore oxidized sugar metabolism of *E. coli garL* knockouts, revealing a novel putative 5-KDG aldolase we named *gudL*. Similarly, *E. clostridioformis* does not contain permeases *gudP* or *garP*, but instead utilizes an ABC transporter within the same operon as a homologous *gudD*. The previously uncharacterized *E. clostridioformis* oxidized sugar transporter *garABC* complements *gudP* and *garP E. coli* knockouts. Based on our discovery that *E. clostridioformis* metabolizes oxidized sugars using a novel combination of homologous genes, convergently evolved enzymes, and alternative transporters, we systematically searched for gut microbial species that contained the *E. coli* or *E. clostridioformis gud/gar* pathway, identifying 887 species with homologous genes, including *Blautia producta* and *Fusobacterium nucleatum*. Thus, we can conclude that oxidized sugar metabolism is far more widespread than previously known and exemplifies the convergent evolution of metabolic adaptation to the inflamed gut.

## Results

### Screening for gut microbial species capable of metabolizing oxidized sugars

We hypothesized the increased abundance of certain gut microbes in IBD may be attributed to their capacity to metabolize oxidized sugars produced during inflammation. To investigate this theory, we screened a selection of species that were both increased in relative abundance during IBD, and lacked a known mechanism behind their increased abundance for growth on oxidized sugars as a carbon source[29].

*E. coli* 5α, *E. clostridioformis* WAL-7855, and *Enterocloster citroniae* WAL-17108 (*Clostridium citroniae*) grew on galactarate as a carbon source, while *Clostridium symbiosum* WAL-14163, *Mediterraneibacter lactaris* CC59-002D (*Ruminococcus lactaris*), and *Bacteroides fragilis* CL03T12C07 did not, despite being associated with IBD (Fig. 2)[29]. The observed growth of these gut microbial species suggests the presence of genes essential for oxidized sugar metabolism. Consequently, we set out to identify these putative metabolic genes.

### Identification of oxidized sugar metabolism gene cluster in *Enterocloster clostridioformis*

We theorized that the observed growth of select gut species on glucarate or galactarate as a carbon source was facilitated by the *gud/gar* pathway previously characterized in *E. coli*. To identify these genes, we searched for homologs using *E. coli* proteins GudD, GarD, GarL, GarR, and GarK as references for the minimal viable enzymatic pathway for oxidized sugar catabolism.

Notably, BLASTp only revealed proteins in *E. clostridioformis* homologous to GudD (percent identity 64.3%, e-value 7.37e-221, bitscore 611), GarD (percent identity 53.9%, e-value 5.11e-191, bitscore 541), GarR (percent identity 67.8%, e-value 2.27e-140, bitscore 395), and two copies of GarK (percent identity 43.5%, e-value 2.14e-98, bitscore 285; percent identity 42.3%, e-value 1.22e-100, bitscore 291). However, these proteins alone are unlikely to fully recapitulate the metabolic pathway to catabolize oxidized sugars, as two key enzymes are missing: Gud/GarP permeases necessary for oxidized sugar import and aldolase GarL, responsible for 5-KDG catabolism to pyruvate and TSA. The missing enzymes raised questions as to how *E. clostridioformis* was able to metabolize galactarate without a complete oxidized sugar catabolic pathway.

Investigating the *E. clostridioformis* genes further, we manually examined the neighborhood of the homologous genes. Interestingly, we found an aldolase that was within the same operon as *garR*, but was not homologous to the *E. coli garL* aldolase. Additionally, while *E. clostridioformis* lacked the *gudP* and *garP* permeases, in the same operon as *gudD*, there was an uncharacterized ABC transporter (Fig. 1B). Given the close proximity of these genes to the homologous *gud/gar* genes, we theorized that the novel aldolase *gudL* and uncharacterized ABC transporter may also be involved in oxidized sugar metabolism.

### Both novel and homologous genes from the *E. clostridioformis gud/gar* gene cluster rescue *E. coli* knockouts

Due to the discovery of novel genes and divergent operon organization, we sought to validate the functionality of the putative *E. clostridioformis gud/gar* gene cluster through the complementation of *E. coli gud/gar* knockouts. Since most *E. coli* K-12 derivatives natively contain the *gud/gar* metabolic pathway, we needed mutant strains of each *gud/gar* gene of interest. For this purpose, we employed the Keio collection, an *E. coli* BW25113 mutant library containing in-frame, single-gene deletions of all nonessential genes, including *gudP, garP, garL*, and *garR*[30].

To create our complementation library, the putative *gud/gar* genes from *E. clostridioformis* were cloned into a pCW-lic vector backbone under the control of an inducible *tac* promoter, creating plasmids to individually express the putative oxidized sugar transporter *garABC*, the putative oxidized sugar aldolase *gudL*, and the homologous reductase *garR* (Supplementary Fig. 1). Then, we transformed each plasmid into the respective Keio *E. coli* knockout strain: pCW-*Clos-gudL* was transformed into Δ*garL E. coli*, pCW-*Clos-garR* into a Δ*garR* mutant, and pCW-*garABC* was transformed into both Δ*gudP*

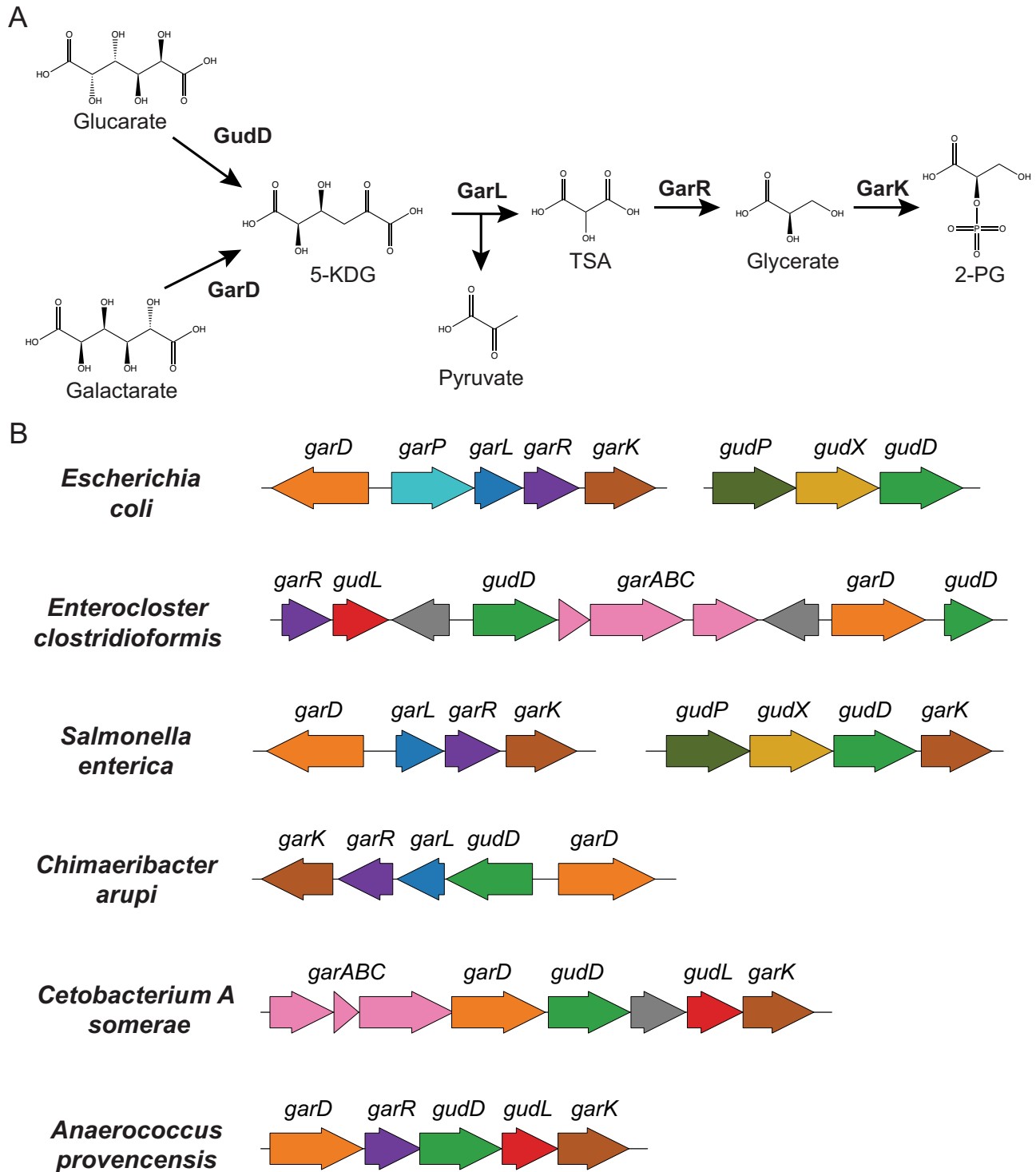

**Fig. 1 | Oxidized sugar metabolism. A** Glucarate and galactarate catabolic pathway. **B** *Gud/gar* gene cluster organization from *E. coli, E. clostridioformis, S. enterica, C. arupi, C. somerae,* and *A. provencensis.*

and Δ*garP* mutants, as we hypothesized that the *E. clostridioformis* ABC transporter would import both glucarate and galactarate.

In order to verify the *E. clostridioformis gud/gar* pathway functionality, we employed a sugar fermentation assay adapted from Faber et al. used to measure fermentation of carbon sources[21]. A minimal peptone fermentation media containing bromothymol blue pH indicator was supplemented with 10 g/L glucose, glucarate, or galactarate as a carbon source[21]. This media changes from a deep blue color to yellow based on the pH change that occurs as bacteria grow and acidify

the media (Fig. 3A). We inoculated *E. coli* into the fermentation media and induced expression with isopropyl β-D-1-thiogalactopyranisode (IPTG). After 48 h, the color change of the media was measured as an indication of carbon source utilization.

Knockouts of *E. coli gudP* and *garL* abolished fermentation of glucarate and galactarate carbon sources, while deletion of *garP* and *garR* attenuated oxidized sugar fermentation (Fig. 3A–G, Supplementary Fig. 2). Complementation of the native *E. coli* gene back into the mutant strain restored utilization of glucarate and galactarate to that

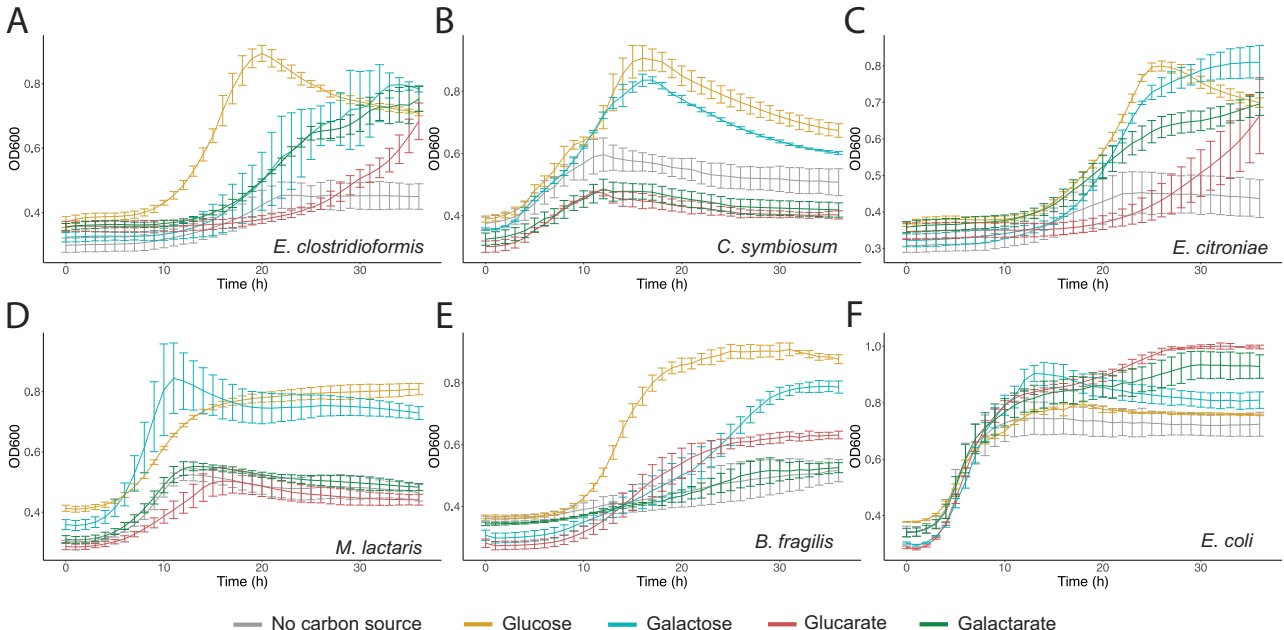

**Fig. 2 | Identification of gut microbial species increased during inflammation capable of metabolizing glucarate or galactarate as a carbon source.** OD600 of **A** *Entercloster clostridioformis* WAL-7855 **B** *Clostridium symbiosum* WAL-14163, **C** *Enterocloster citroniae* WAL-17108, **D** *Mediterraneibacter lactaris* CC59-002D, **E** *Bacteroides fragilis* CL03T12C07, and **F** *E. coli* 5α grown in media with glucose, galactose, glucarate, or galactarate as an additional carbon source (*n* = 3 biological replicates). Data are presented as the mean ± standard deviation. Source data are provided as a Source Data file.

of the wild-type *E. coli* in all cases, validating the ectopic expression methodology (Fig. 3A–G, Supplementary Fig. 3).

Complementation of the Δ*garP* mutant with *E. clostridioformis garABC* rescued the knockout, while the same genes in Δ*gudP* mutants only partially rescued the metabolic defects (Fig. 3B, E). Complementation of Δ*garR* mutants with the homologous *E. clostridioformis garR* fully recovered the oxidized sugar metabolism phenotype on both glucarate and galactarate. Most notably, cross-species complementation of the *E. coli* Δ*garL* mutants with the putative *E. clostridioformis* aldolase *gudL* restored fermentation of glucarate and galactarate as strongly as isogenic complementation (Fig. 3A, C, F).

The restoration of *E. coli* knockout fermentation of oxidized sugars by heterologous expression of *E. clostridioformis* genes *garABC*, *gudL*, and *garR* demonstrates the functional equivalency of the encoded enzymes in the bacterial metabolism of oxidized sugars.

**The *dapA*-like gene *gudL* encodes a putative 5-KDG aldolase**

The discovery of the novel GudL aldolase in *E. clostridioformis*, which catalyzes the same function as GarL in *E. coli*, prompted us to study its evolutionary history. Thus, we investigated the evolution of GudL to gain insights into its emergence and diversification across bacterial species. Despite the low amino acid pairwise identity of 7% between the *E. coli* 5-KDG aldolase GarL and *E. clostridioformis* aldolase GudL, their shared functionality led to our hypothesis that this similarity is likely due to convergent evolution. An Interproscan search revealed that the *E. clostridioformis* GudL is annotated as "DapA-like" (IPR017648) and shares no common protein domain annotations with the *E. coli* 5-KDG aldolase GarL, substantiating the convergent evolution hypothesis (Fig. 4A, B)[31].

We obtained further evidence supporting different common ancestors through structural comparisons of GarL and GudL. The Alphafold-predicted structures of GudL and the X-ray crystallography structure of *E. coli* GarL (PDB: 1DXE) exhibit a root mean square deviation (RMSD) of 3.05, indicating structural divergence (Fig. 4C, D)[32]. In contrast, as previously mentioned, the *E. clostridioformis* GarD, GudD, and GarR exhibit high shared pairwise amino

acid identity to their *E. coli* counterparts (57%, 64%, and 68%, respectively), implying homology.

Together, these data suggest the *E. clostridioformis* gud/gar pathway consists of GudD, GarD and GarR enzymes homologous to their *E. coli* counterparts, but uses a newly discovered, convergently evolved aldolase GudL predicted to catalyze the conversion of 5-KDG to TSA and pyruvate, and an ABC transporter that functionally compensates for the absence of GudP or GarP to facilitate galactarate import (Fig. 1B).

**Delineation of the putative GudL clade**

To delineate GudL from other DapA-like proteins, we used a combination of phylogenetic analyses, structural prediction, and sequence conservation, identifying key residues that differentiate GudL from DapA. Our phylogenetic analysis of DapA-like proteins revealed a putative GudL clade, based on the positions of the *E. clostridioformis* GudL and the branch length of the clade (Fig. 5A).

Ancestral sequence reconstruction showed changes in the catalytic residues from the ancestral state (node 2) (Fig. 5B). Notably, there was a predicted change from Y110 to M110 and N194 to K194 from the ancestral state to the GudL clade, which is universally conserved among the putative GudL clade. These residues unique to the GudL clade may contribute to its function, and delineate it from other DapA-like enzymes. We mutated the *E. clostridioformis* GudL sequence to change position M110 and K194 to a neutral amino acid with no side chain (M110A, K194A), and to their ancestral states (M110Y, K194N). Mutating these residues to alanine abolished the fermentation of both glucarate and galactarate, with the exception of K194A, which reduced but did not abolish oxidized sugar metabolism (Fig. 5D, E). These results indicate that the GudL clade can be delineated from other DapA at node 1.

We identified putative catalytic residues of GudL through structural alignments with the homologous 4-hydroxy-tetrahydrodipicolinate synthase (3PB2), revealing similar positioning of K167 to the DapA catalytic residue K161 (Fig. 5C). In DapA, K161 catalyzes Schiff base formation during the synthesis of 2,3-

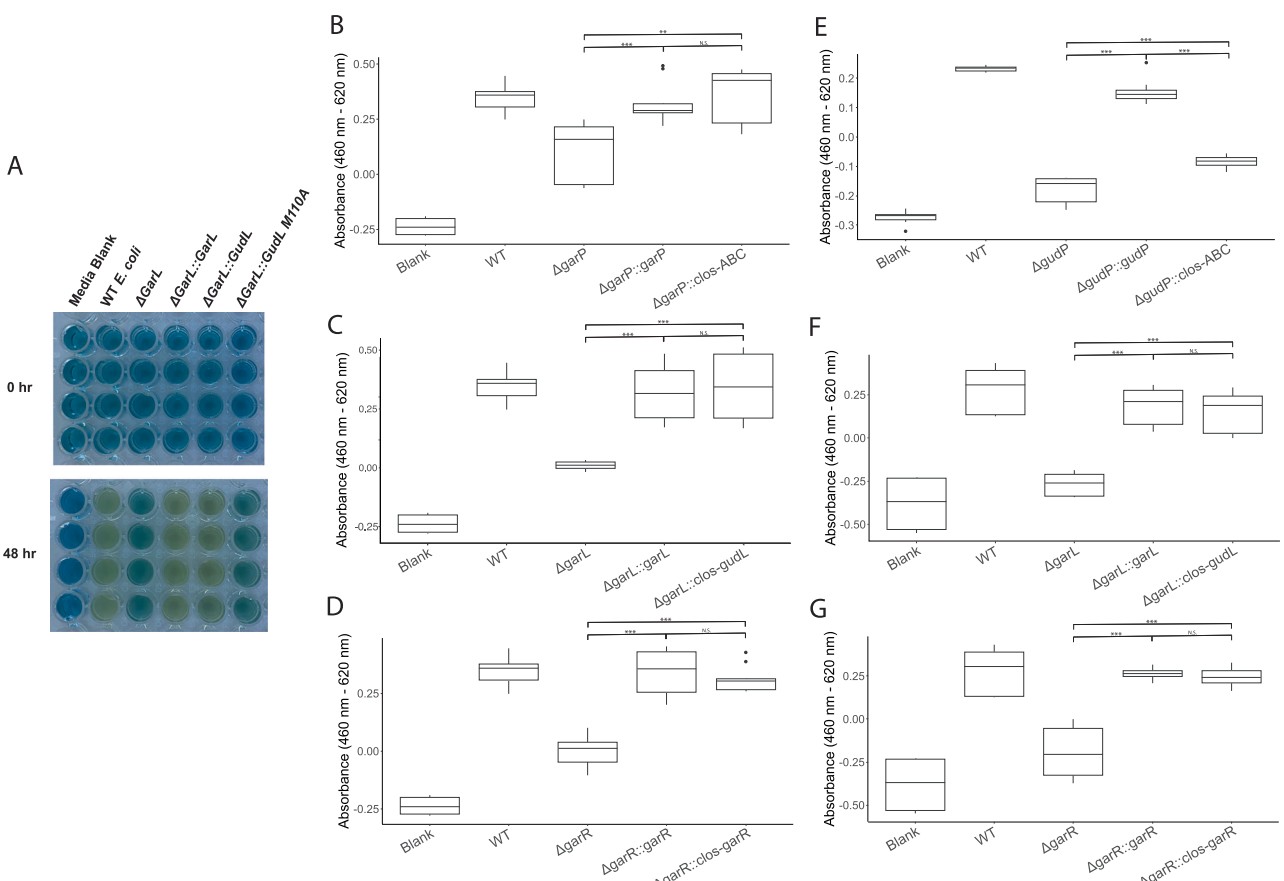

**Fig. 3 | Complement metabolism of oxidized sugars. A** Representative images of fermentation media with 10 g/L galactarate inoculated with *E. coli* BL21, *ΔgarL*, *ΔgarL::garL*, *ΔgarL::Clos-gudL* or *ΔgarL::gudL-M110A* at 25 °C for 48 h. For the full plate image, see Supplemental Fig. 2. **B** Fermentation assay after 48 h of *ΔgarP*, *ΔgarP::garP*, and *ΔgarP::Clos-ABC E. coli* on galactarate as a carbon source (*n* = 9 biological replicates. *ΔgarP::garP p* = 9.7e-04, *ΔgarP::Clos-ABC p* = 0.0006, *ΔgarP::-garP* to *ΔgarP::Clos-ABC p* = 0.55). **C** Fermentation assay after 48 h of *ΔgarL*, *ΔgarL::garL*, and *ΔgarL::Clos-gudL E. coli* on galactarate as a carbon source (*n* = 9 biological replicates. *ΔgarL::garL p* = 0.002, *ΔgarL::Clos-gudL p* = 0.0036, *ΔgarL::-garL* to *ΔgarL::Clos-gudL p* = 0.77). **D** Fermentation assay after 48 h of *ΔgarR*, *ΔgarR::garR*, and *ΔgarR::garR E. coli* on galactarate as a carbon source (*n* = 9 biological replicates. *ΔgarR::garR p* = 1.6e-07, *ΔgarP::Clos-garR p* = 1.9e-09, *ΔgarR::-garR* to *ΔgarP::Clos-garR p* = 0.41). **E** Fermentation assay after 48 h of *ΔgudP*, *ΔgudP::gudP*, and *ΔgudP::Clos-ABC E. coli* on glucarate as a carbon source (*n* = 8

biological replicates. *ΔgudP::gudP p* = 4.8e-10, *ΔgudP::Clos-ABC p* = 2.8e-04, *ΔgudP::gudP* to *ΔgudP::Clos-ABC p* = 8e-08). **F** Fermentation assay of *ΔgarL*, *ΔgarL::garL*, and *ΔgarL::Clos-garL E. coli* on glucarate as a carbon source (*n* = 8 biological replicates. *ΔgarL::garL p* = 6.4e-07, *ΔgarL::Clos-garL p* = 3e-06, *ΔgarL::-garL* to *ΔgarL::Clos-garL p* = 0.59). **G** Fermentation assay of *ΔgarR*, *ΔgarR::garR*, and *ΔgarR::Clos-garR E. coli* on glucarate as a carbon source (*n* = 8 biological replicates. *ΔgarR::garR p* = 3.9e-05, *ΔgarR::Clos-garR p* = 3.4e-05, *ΔgarR::garR* to *ΔgarR::Clos-garR p* = 0.4). Wild-type *E. coli* BL21 was used as a positive control, and a media blank as a negative control. N.S *p* > 0.05. *\*p* < 0.05. *\*\*p* < 0.01. *\*\*\*p* < 0.005. Two-sided *t*-tests were used without adjustment for multiple comparisons. The rectangles represent the interquartile range (IQR; 25th to the 75th percentile). The center line of each box shows the median. The whiskers reach from the lower quartile minus 1.5 times the IQR, to the upper quartile plus 1.5 times the IQR. Source data are provided as a Source Data file.

dihydrodipicolinate from pyruvate and L-aspartate-4-semialdehyde[33]. Consequently, we hypothesize that K167 may play an important role in the predicted catalytic reaction of 5-KDG by GudL. To verify the importance of K167, we mutated the GudL sequence to change position K167 into alanine. The mutation abolished the utilization of both glucarate and galactarate (Fig. 5D, E).

### Oxidized sugar metabolism is a strain-specific capability widely distributed across the microbiome

Our original hypothesis was that commensal microbes may have adapted to utilize oxidized sugars in order to gain a competitive advantage in the inflamed gut. Once we demonstrated through genetic complementation that the divergent pathway for oxidized sugar metabolism in *E. clostridioformis* was functional, we expanded our search to other IBD-associated taxa and commensal microbes. Using ProkFunFind, a tool to annotate genes across gut microbial species, we searched the 85,202 prokaryotic genomes from the Genome Taxonomy Database (GTDB) for species that contained either homologs to

the *E. coli* GudD/GarD, GarL, and GarR proteins, or homologs to *E. clostridioformis* GudD/GarD, GudL, and GarR[34,35]. Species that contained GudD or GarD, GarL or GudL, and GarR were considered to contain the minimum viable *gud/gar* pathway. GarK was excluded from the minimum viable pathway, as upon constructing the *garK* gene tree, we found that *garK* was polyphyletic, suggesting independent recruitment of kinases multiple times throughout evolution. This pattern contrasts sharply with the other *gud/gar* genes (*gudD/garD*, *gudL/garL*, and *garR*), which show consistent co-evolution as a single functional unit (Supplementary Fig. 3). This method identified the putative minimum viable *gud/gar* pathway in 887 gut microbial species, including *E. citroniae*, and *Fusobacterium nucleatum E* (Fig. 6). *Gud/gar* was primarily found in the phyla Bacillota (316 species), Pseudomonadota/Proteobacteria (505 species), and Fusobacteriota (10 species). Interestingly, GarL was primarily found in Pseudomonadota, while GudL is widely spread among a diverse range of phyla, including Bacillota and Fusobacteriota (Fig. 4E, F).

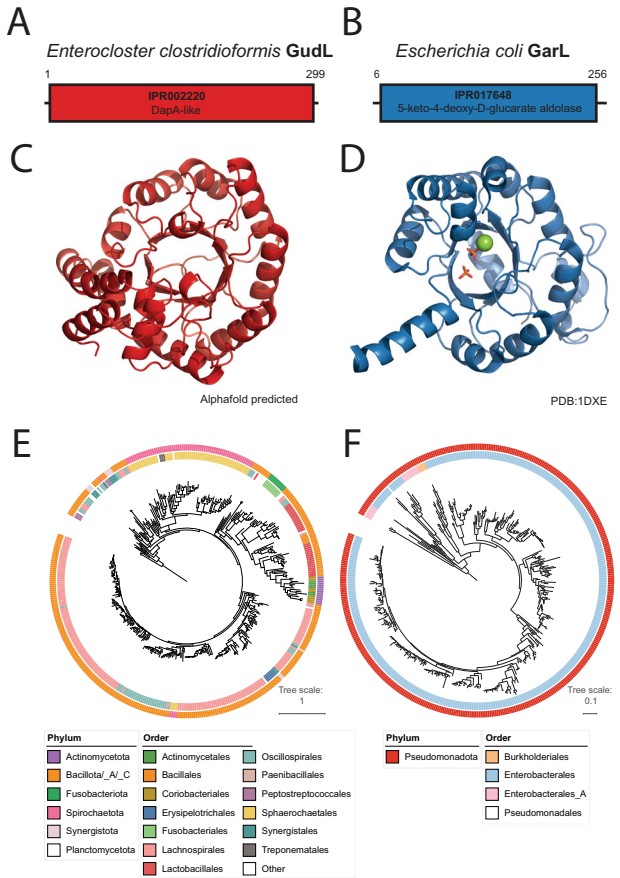

**Fig. 4 | Comparison of GudL and GarL.** Interproscan annotations of **A** *E. clostridioformis* GudL and **B** *E. coli* GarL. **C** Alphafold-predicted structures of *E. clostridioformis* GudL and **D** the monomer X-ray diffraction structure *E. coli* GarL obtained from the RCSB Protein Data Bank (ID: 1DXE). The RMSD of these two structures was 3.05, calculated using TM-align[51,56]. Phylogenetic gene trees of **E** *E. clostridioformis* GudL and **F** *E. coli* GarL.

Oxidized sugar metabolism appears to be a strain-specific function. For example, 2 out of 38 *Fusobacterium polymorphum* genomes, and 6 out of 18 *Fusobacterium mortiferum* genomes contain the *gud/gar* pathway (Supplementary Fig. 4).

### Horizontal gene transfer of the *gud/gar* gene cluster from Bacillota to Fusobacteriota

Given that the gud/gar pathway was present in divergent phyla such as Bacillota, Fusobacteriota, and Pseudomonadota, we sought to understand the evolutionary history of the *gud/gar* pathway by reconstructing phylogenetic trees of the GudD, GarD, GudL, and GarR genes of the 887 species with the gud/gar pathway. The incongruence of the species cladogram and the GarD, GudD, and GarR gene trees suggest that the gene cluster may have spread via horizontal gene transfer, notably indicating a potential transfer from Bacillota to Fusobacteriota, possibly from the *Enterococcus* spp. clade. The GarD, GudD, and GarR gene trees show the Fusobacteriota clade nested within the Bacillota clade, despite being from a different phyla (Fig. 7). The similarity between the *Enterococcus* spp. GarR to the *Fusobacterium* spp. GarR in both sequence identity and gene cluster organization suggests that a horizontal gene transfer event of the *garR* gene cluster occurred between Bacillota and Fusobacteriota (Fig. 7). For example, the similarity of the GarR sequences between *Fusobacterium B.* sp. and *Enterococcus D. casseliflavus* is 82%. Furthermore, *Fusobacterium B.* sp. and *Enterococcus D. casseliflavus* share 5 genes in the *garR* gene

clusters, these being *gudD*, *gudL*, *garK*, *garR*, and and *cDaR*, implicating horizontal gene transfer. This suggests that Fusobacteriota horizontally acquired a gene cluster that could be adaptive for inflammatory environments, potentially contributing a mechanism to the previously observed association of Fusobacteriota with IBD and colorectal cancer[36].

To verify the functionality of the *Fusobacterium nucleatum gud/gar* genes, we complemented Keio *E. coli* knockouts with the *F. nucleatum* genes *gudL* and *garR*, which partially restored fermentation of both glucarate and galactarate (Fig. 7C, D).

### IBD patients exhibit an increased relative abundance of the *gud/gar* pathway in fecal metagenomics and metatranscriptomics

We hypothesized that the ability to utilize oxidized sugars was an adaptation that conferred a competitive advantage to microbes in the inflamed gut. Therefore, we sought to determine if the relative abundance of microbes encoding the gud/gar pathway was increased in IBD compared to non-inflamed controls. However, due to our previous discovery that *gud/gar* are accessory genes, we mapped metagenomic reads directly to a database of gud/gar operons rather than examine species-level changes. Using the homologous *gud/gar* genes *garD*, *gudD*, and *garR* as a reference database, we mapped the stool metagenomic reads from 1261 metagenomes of Human Microbiome Project 2 (HMP2) data (UC $n = 337$, CD $n = 565$, non-IBD $n = 359$) and determined that the genes in the *gud/gar* pathway are found in higher relative abundance in IBD patients compared to non-IBD patients (Mann U Whitney test, CD $p = 6.181e-08$ $W = 122,824$, UC $p = 0.01763$ $W = 54,200$, UC-CD $p = 0.008988$ $W = 105,092$) (Fig. 8A).

We further investigated whether these genes were being expressed in vivo. Stool metatranscriptomic analysis of 1106 metatranscriptomes (UC $n = 184$, CD $n = 292$, non-IBD $n = 630$) found an increased relative abundance of *gud/gar* pathway transcripts in IBD patients (MTX: CD $p = 0.002261$ $W = 103,445$, UC $p = 0.07443$ $W = 52,967$, UC-CD $p = 0.3782$ $W = 28,152$), suggesting that not only that the relative abundance of these species is increased during IBD, but they are also actively transcribing the *gud/gar* genes suggesting their functional utility (Fig. 8C).

However, we wanted to confirm whether the alternative pathway identified in *E. clostridioformis* was contributing to the increase in *gud/gar* in IBD patients (UC $n = 166$, CD $n = 270$, non-IBD $n = 570$). To do this, we used 380 putative *gudL* homologs as a reference database and determined that *gudL* was increased in patients with IBD for both the metatranscriptomes and metagenomes (Fig. 8B, D) (Mann–Whitney U Test, MGX: CD $p = 3.331e-10$ $W = 126,260$, UC $p = 0.0002354$ $W = 50,742$, UC-CD $p = 0.008988$ $W = 105,092$, MTX: CD $p < 2.2e-16$ $W = 123,662$, UC $p = 1.614e-14$ $W = 38,031$, UC-CD $p = 0.7831$ $W = 27,261$). When we examined changes in *garL* between IBD patients and non-IBD individuals, we found that *garL* genes and transcripts were increased in CD patients, but not in UC patients compared to healthy controls (Supplementary Fig. 5) (Mann–Whitney U Test, MGX: CD $p = 8.808e-05$ $W = 115,671$, UC $p = 0.07631$ $W = 56,227$, UC-CD $p = 0.03518$ $W = 102,641$, MTX: CD $p = 1.752e-08$ $W = 99,977$, UC $p = 0.2358$ $W = 57,046$, UC-CD $p = 0.005956$ $W = 28,776$).

## Discussion

In this study, we have shown that disparate taxa recapitulate the metabolism of glucarate and galactarate utilizing enzymatically divergent, yet functionally equivalent, *gud/gar* pathways. We identified this alternative enzymatic pathway in *E. clostridioformis* and demonstrated that the homologous reductase *garR*, uncharacterized transporter *garABC*, and putative 5-KDG aldolase *gudL* are able to complement the relevant *E. coli gud/gar* knockouts. Based on the discovery of this parallel pathway, we systematically searched for commensal species with the *gud/gar* pathway and identified 887 species, spanning multiple phyla, with the minimum viable pathway. From this,

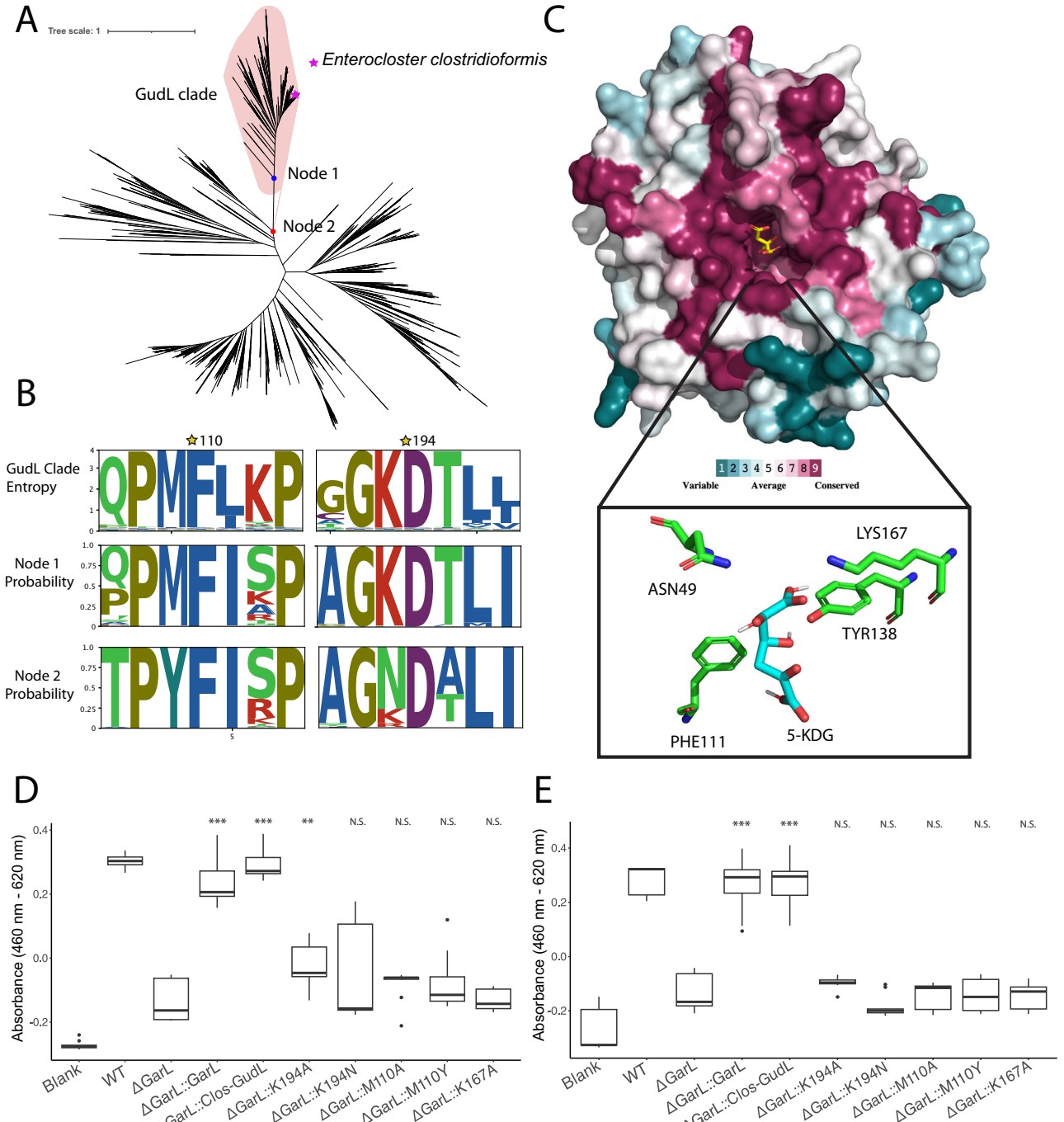

**Fig. 5 | Delineation of GudL. A** Gene tree constructed from putative GudL sequences and related enzymes annotated as dapA-like (COG0329), showing a possible delineation of the GudL clade and the location of *E. clostridioformis* GudL. **B** Diagram showing the conservation (entropy) of active site residues in clade 1, as well as the GRASP predicted ancestral states of Node 1 (GudL clade) and Node 2 (putative ancestral node). Active site residues with a predicted change from the ancestral state are labeled with a star. **C** AlphaFold-predicted structure colored by the degree of conservation of amino acid residues based on a ConSurf analysis. Red represents conserved regions, and green represents variable regions. A docked 5-KDG is shown on the structure (yellow and blue). The potential active site residue interactions with 5-KDG are shown, as predicted by Autodock vina. **D** Fermentation assay after 48 h of wild-type *E. coli BL21*, *ΔgarL*, *ΔgarL::garL*, *ΔgarL::Clos-GudL E. coli*,

and *gudL* mutants K194A, K194N, M110A, M110Y, and K167A on galactarate (*ΔgarL::garL p = 5.8e-09, ΔgarL::Clos-GudL p = 8.2e-11*, K194A *p = 0.0021*, K194N *p = 0.18*, M110A *p = 0.062*, M110Y *p = 0.085*, K167A *p = 0.79*) and **E** glucarate (*ΔgarL::garL p = 7.2e-07, ΔgarL::Clos-GudL p = 8e-08*, K194A *p = 0.099*, K194N *p = 0.093*, M110A *p = 0.73*, M110Y *p = 0.82*, K167A *p = 0.65*) as a carbon source (*n = 9* biological replicates). Significance is relative to *ΔgarL*. Experimental controls are shared in Fig. 7D. N.S *p > 0.05*. *\*p < 0.05*. *\*\*p < 0.01*. *\*\*\*p < 0.005*. Two-sided *t*-tests were used without adjustment for multiple comparisons. The rectangles represent the IQR. The center line of each box shows the median. The whiskers reach from the lower quartile minus 1.5 times the IQR, to the upper quartile plus 1.5 times the IQR. Source data are provided as a Source Data file.

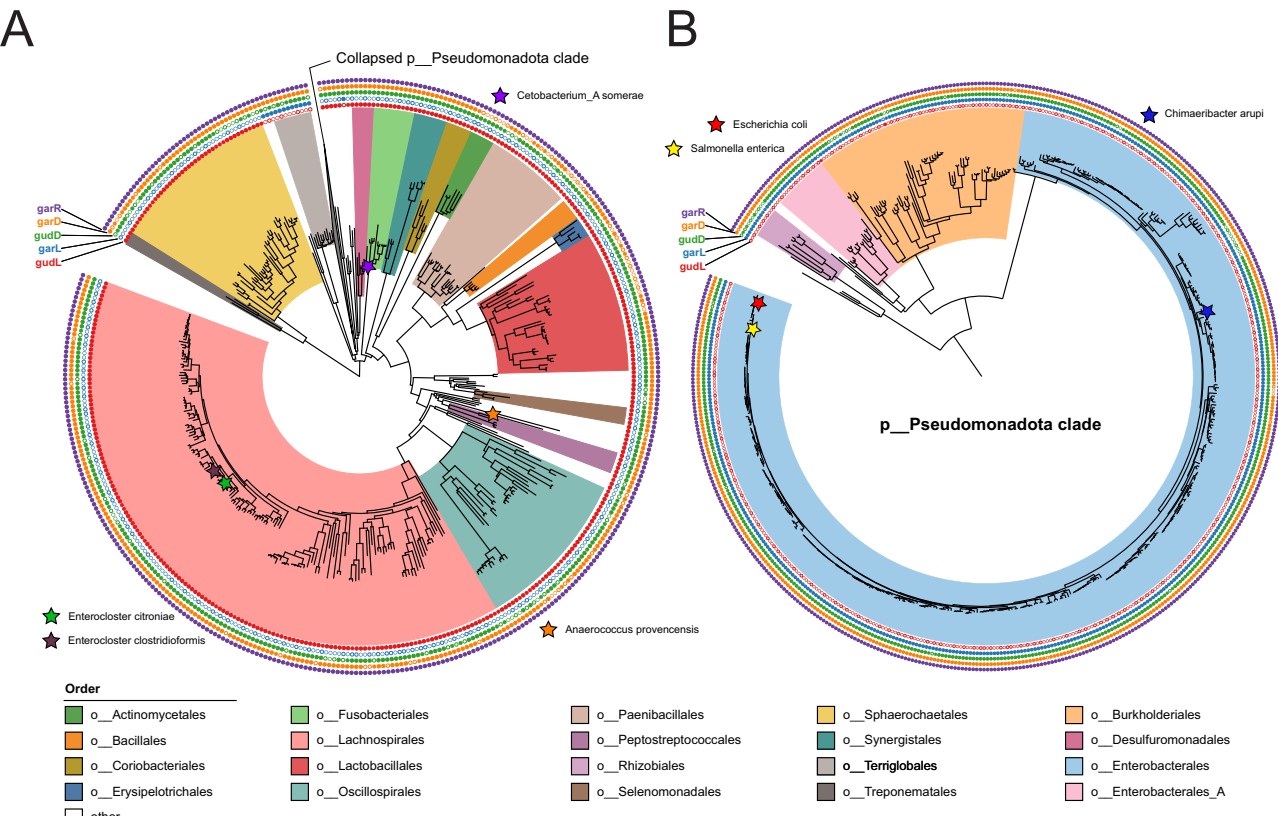

**Fig. 6 | Species tree of GTDB taxa.** The minimum viable gud/gar pathway is annotated with colored dots showing the presence or absence of GarR, GarD, GudD, and GudL in each species. The tree is divided into two segments, with one section collapsed in each part. **A** Section of the tree showing primarily species with GudL instead of GarL, including phyla such as Bacillota and Fusobacteriota. **B** Subsection of the Pseudomonadota clade, showing primarily species with GarL instead of GudL. The species tree was generated by pruning the GTDB species tree using the gotree prune command.

we can conclude that the *gud/gar* pathway, which was previously considered a pathogen colonization factor, is present in many commensals, often via the alternative enzymatic pathway seen in *E. clostridioformis*.

Functional complementation of *E. coli* knockouts by the divergent *E. clostridioformis* genes reveals intriguing gaps in our understanding of oxidized sugar metabolism. *E. coli* Δ*garP* mutants show attenuated, rather than abolished, metabolism of galactarate, implying alternative methods of galactarate transport. GudP may be capable of importing galactarate given the structural similarity and the known promiscuity of GudD to dehydrate both glucarate and galactarate[27]. Furthermore, the *E. clostridioformis garABC* rescues both Δ*gudP* and Δ*garP* mutants, suggesting a single ABC transporter mediates glucarate and galactarate uptake in *E. clostridioformis*, unlike the unique *E. coli* symporters[28].

Of particular note is our discovery of the novel *E. clostridioformis* aldolase GudL, which shares a function with the *E. coli* aldolase GarL, even though they do not share a recent common ancestor. Complementation of an *E. coli* Δ*garL* knockout with *E. clostridioformis gudL* can fully rescue the defects in oxidized sugar metabolism, implicating *gudL* as a putative novel 5-KDG aldolase. However, the divergent structures of GudL and GarL, along with a lack of significant sequence similarity, suggest that they likely do not share a recent common ancestor. The presence of similar catalytic activities in two evolutionarily unrelated aldolases reflects a potential case of convergent evolution. Previous studies have documented cases of convergent evolution in aldolases, such as L- and D- threonine aldolases, across various bacterial species[37].

Interestingly, although GudL and GarL are distinct, *E. coli* and *E. clostridioformis gud/gar* enzymes GudD, GarD, and GarR share high sequence similarity. We speculate that there could be horizontal gene transfer of these *gud/gar* genes between Bacillota and Pseudomonadota, although the directionality is unclear. However, our phylogenetic and gene cluster analyses do provide evidence for the direction of the gut adaptive horizontal gene transfer of the *gud/gar* pathway between Fusobacteriota and Bacillota. The *garR* and *garD* gene trees show the Fusobacteriota clade nested within the Bacillota clade. Combined with our synteny analyses, this suggests that Fusobacteriota acquired the *gud/gar* pathway from Bacillota. Past literature shows that horizontal gene transfer between *Fusobacterium nucleatum* and Bacillota is prevalent, and can be attributed to their close proximity in the oral cavity[38,39]. Inflammation and oxidative stress are reported to be involved in the initiation of colorectal cancer, and IBD patients are more likely to get colorectal cancer[40]. Given this, the potential involvement of *Fusobacterium* in sugar oxidation is an important finding, as the presence of *Fusobacterium* in the gut is correlated with colon cancer and increased in patients with Crohn's disease[36,41,42].

Between the ABC transporter and novel *gudL* aldolase, we observe that divergent phyla have recapitulated the same function using alternative enzymes. Using the different operon organization and non-homologous genes identified in *E. clostridioformis* and *E. coli*, we have expanded the annotation of a function to include 887 additional gut microbial species. However, given the non-homologous aldolase we identified, it is possible that there are additional pathways for the metabolism of oxidized sugars that were not found during our search.

However, the strain-specificity of *gud/gar* introduces complexities when evaluating contributions to shifts in the microbial community. Taxonomic profiling of the microbiome has, at best, species-level resolution, precluding definitive conclusions about whether the enriched strains contain *gud/gar* and, therefore, metabolize oxidized sugars. Similarly, though *gud/gar* transcripts were increased in IBD,

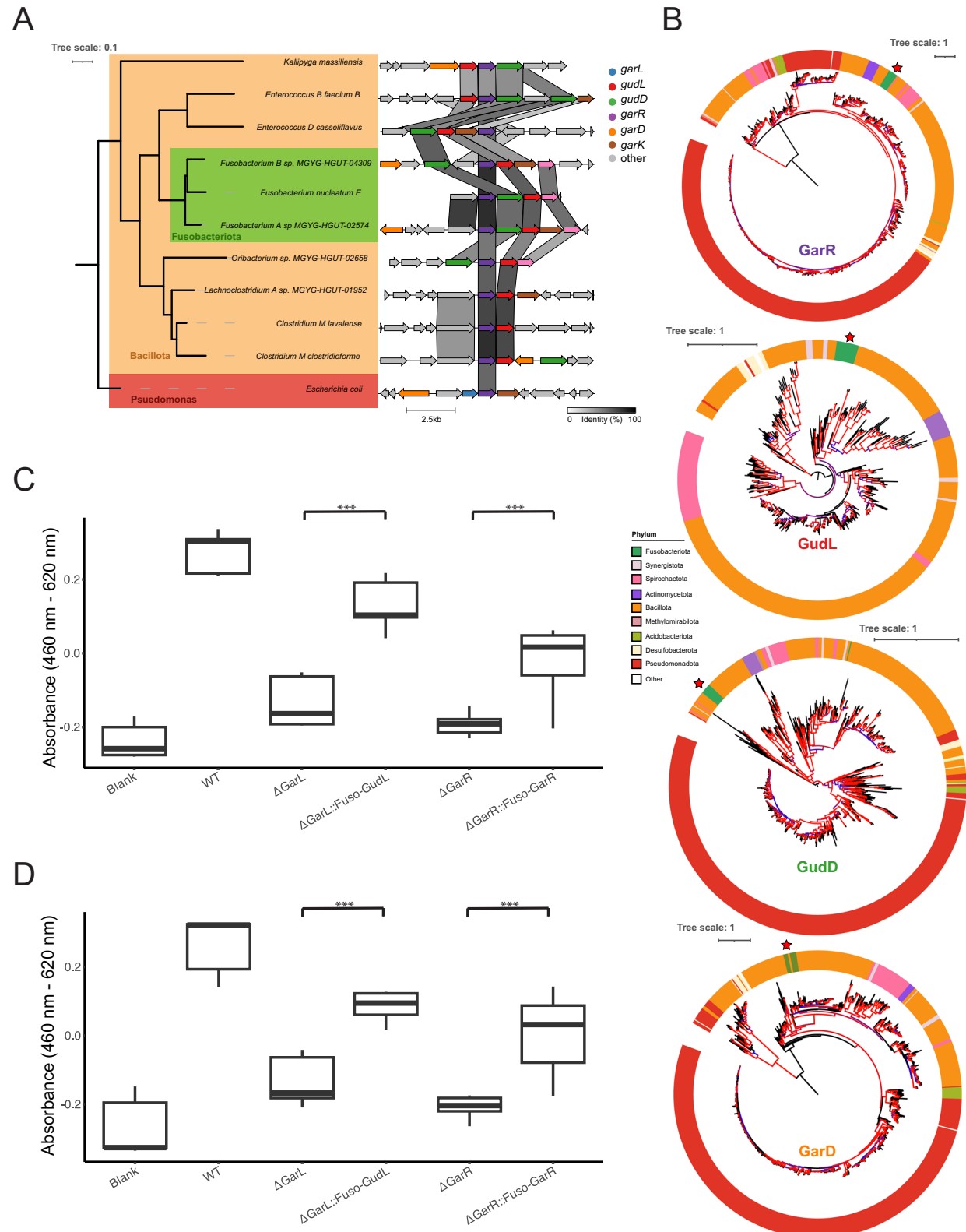

limited read coverage prevented the evaluation of changes in transcriptional regulation[43]. The regulation of *gud/gar* transcription during inflammation, particularly regulatory changes between flares and periods of remission, remains an open avenue of investigation.

Though persisting knowledge gaps hinder our comprehensive understanding of how *gud/gar* contributes to inflammation-driven perturbations of the gut microbiome, utilization of alternative carbon sources via oxidized sugar metabolism offers additional insight into potential mechanisms underpinning the increased abundance of several obligate anaerobic taxa. Many obligate anaerobes are decreased in IBD due to a lack of mechanisms to detoxify the reactive oxygen and nitrogen species produced during inflammation[44–46]. Other genera benefit from the increased oxidative and nitrosative stress[12–14]. However, *Blautia* and *Lachnoclostridium* are obligate anaerobes enriched

**Fig. 7 | Oxidized sugar metabolism in Fusobacteriota. A** Phylogenetic tree and synteny analysis of the *garR* and *gudL* cluster in representative *Fusobacterium* spp. and Bacillota species. Homologous genes are represented by the same color, and darker links show higher similarity between homologous genes. Diagram produced by clinker v0.0.28. The phylogenetic tree was generated using IQ-TREE2 from the putative GarR proteins of the representative Bacillota and *Fusobacterium* spp., with *Escherichia coli* GarR as the outgroup. **B** Gene trees of GarR, GudL, GarD, and GudD in gut microbial proteome. Genomes from GTDB[34]. The trees are colored by phylum. Branches are colored to reflect bootstrap values, ranging from 75 (blue) to 100

(red). **C** Fermentation assay after 48 h of wild-type *E. coli*, *ΔgarL*, *ΔgarL::Fuso-gudL*, *ΔgarR*, and *ΔgarR::Fuso-garR E. coli* on galactarate (*ΔgarL::Fuso-gudL p = 9.5e-08*, *ΔgarR::Fuso-garR p = 8.2e-04*) and **D** glucarate (*ΔgarL::Fuso-gudL p = 6e-07, ΔgarR::-Fuso-garR p = 6.5e-04*) as a carbon source (*n = 9* biological replicates). N.S *p > 0.05*. **p < 0.05*. ***p < 0.01*. ****p < 0.005*. Two-sided *t*-tests were used without adjustment for multiple comparisons. The rectangles represent the IQR. The center line of each box shows the median. The whiskers reach from the lower quartile minus 1.5 times the IQR, to the upper quartile plus 1.5 times the IQR. Source data are provided as a Source Data file.

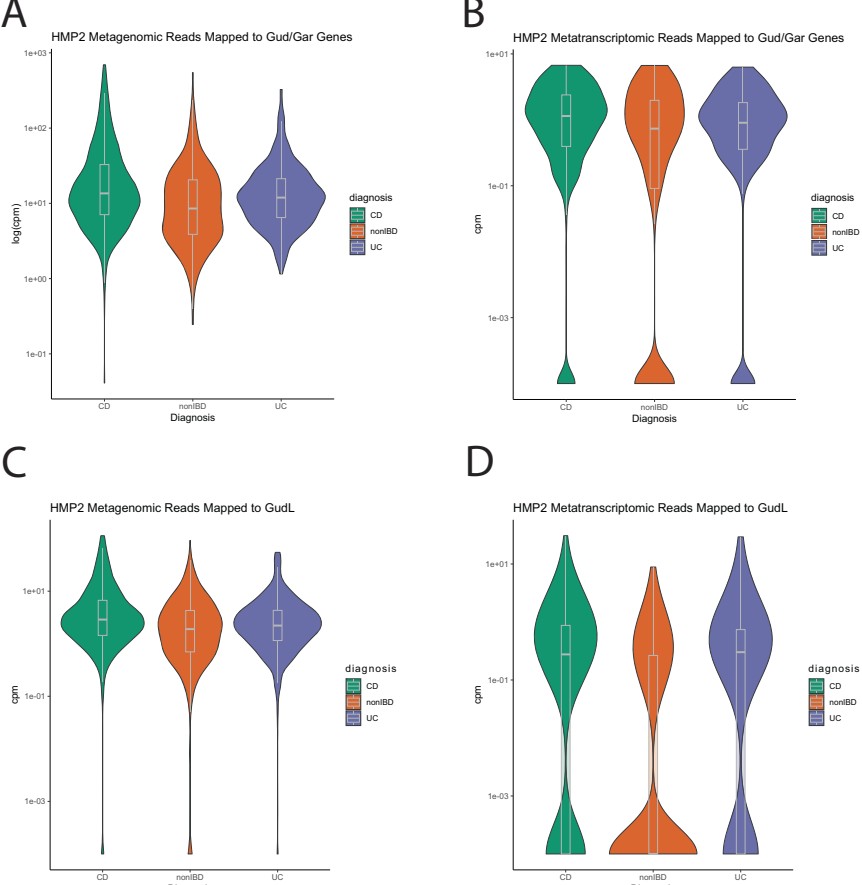

**Fig. 8 | Metagenomic and metatranscriptomic analysis of *gud/gar* and *gudL* in IBD and non-IBD patients. A** Metagenomic (UC *n = 337*, CD *n = 565*, non-IBD *n = 359*) analyses of *gudD, garD, garL*, and *garR* (Two-sided Mann–Whitney U Test, CD *p = 3.331e-10 W = 126,260*, UC *p = 0.0002354 W = 50,742*, UC-CD *p = 0.008988 W = 105,092*) and **B** *gudL* only (Two-sided Mann–Whitney U Test, CD *p = 1.107e-09 W = 125,511*, UC *p = 0.01401 W = 53,978*, UC-CD *p = 0.0002738 W = 108,977*). **C** Metatranscriptomic analysis of genes *gudD, garD, garL, gudL*, and *garR* in UC, CD,

or non-IBD patients (UC *n = 166*, CD *n = 270*, non-IBD *n = 570*) (Two-sided Mann–Whitney U Test, CD *p = 0.002261 W = 103,445*, UC *p = 0.07443 W = 52,967*, UC-CD *p = 0.3782 W = 28,152*). **D** Metatranscriptomic analyses of *gudL* in UC, CD, or non-IBD patients (UC *n = 166*, CD *n = 270*, non-IBD *n = 570*) (Two-sided Mann–Whitney U Test, CD *p = 1.548e-19 W = 123,662*, UC *p = 1.614e-14 W = 38,031*, UC-CD *p = 0.7831 W = 27261*). Data is presented as violin plots with lines representing the median and IQR. Data from HMP2 and HPFS[16].

during inflammation that cannot utilize nitrates as electron acceptors, deviating from the pattern of decreased abundance seen in most obligate anaerobes and leaving the mechanism behind their increased abundance a mystery[8].

These phenomena suggest additional mechanisms at work behind the inflammation-induced microbial perturbations. Here, we provide a potential mechanism through which *Blautia* and *Lachnoclostridium* spp., alongside many other commensal species, are able to metabolize the oxidized sugars produced in the inflamed gut as orthogonal carbon sources and demonstrate that the *gud/gar* genes providing this functionality are increased in abundance and transcription in IBD patients compared to non-IBD controls. Based on this evidence, we propose that the metabolism of oxidized sugars, which was previously

considered a pathogen colonization factor, may also be one of the many metabolic adaptations by commensal species to the inflamed gut. Our findings implicate oxidized sugar metabolism as a potential mechanism contributing to inflammation-related gut dysbiosis and shed new light on microbial adaptation to the inflamed gut.

## Methods
### Growth of anaerobic microbes
Bacterial strains were obtained from the NIH Biodefense and Emerging Infections Research Resources Repository (BEI). Strains were inoculated from a glycerol stock and grown under anaerobic conditions (90% $N_2$, 5% $CO_2$, 5% $H_2$) at 37 °C in an anaerobic chamber (Coy Laboratory Products). Strains were grown in minimal brain–heart

infusion (BHI) broth (Research Products International, B11000) without glucose, and supplemented with 2 g/L glucose or galactarate.

### Fermentation assay

Following the method provided by Faber et al., the fermentation media consisted of 10 g/L peptone (Criterion, C6641), 10 g/L carbon source (no carbon source, glucose, glucarate, or galactarate), and 24 mg/L bromothymol blue (Sigma-Aldrich, 114421)[21]. The media was corrected to a pH of 7.6 and autoclaved to sterilize. Transformed *E. coli* were shaken in LB media supplemented with 100 µg/mL carbenicillin (GoldBio, 00901C103) overnight at 37 °C. 20 µL of the overnight culture was inoculated into 1 mL of fermentation media supplemented with 100 µM isopropyl β-d-1-thiogalactopyranoside (IPTG, GoldBio, 221105I2481) and 100 µg/mL carbenicillin. 200 µL triplicates or quadruplicates were pipetted into a clear-bottom 96-well acrylic plate (Celltreat, 229592), sealed with a Breathe Easy membrane (Electron Microscopy Sciences, 70536), and incubated at 25 °C for 48 h. The endpoint absorbance at 460 nm and 620 nm was read with a SpectraMax M5 plate reader and subtracted as a reference wavelength.

### Construct development

**pCW-Clos constructs.** To ectopically express *garABC, garL*, and *garR* from *E. clostridioformis* in Keio *E. coli* mutants, the gene(s) of interest were amplified and inserted into a pCW-lic vector backbone (Addgene plasmid #26098; http://n2t.net/addgene:26098; RRID: Addgene_26098). A polymerase chain reaction was performed on *E. clostridioformis* genomic DNA via polymerase chain reaction (PCR) using Phusion High Fidelity DNA Polymerase (NEB, M0530S) and the primers listed in Supplementary Table 2. The product was purified using a Monarch PCR & DNA Cleanup Kit (NEB, T1030S). The pCW-lic backbone was cut with restriction enzymes NdeI (NEB, R0111S) and KpnI-HF (NEB, R3142S) and purified with a Monarch PCR & DNA Cleanup Kit. A Gibson assembly was performed with Gibson Assembly Master Mix (NEB, E2611S) according to the manufacturer's protocols and stored at −20 °C until use.

**pCW-Keio constructs.** The gene of interest (*gudP, garP, garL*, or *GarR)* was amplified off of a wild-type Keio *E. coli* genome via and ligated into the pCW-lic vector using the same protocol as noted above, with primers listed in Supplementary Table 2.

**pCW-Fuso constructs.** The genes of interest (*gudL, garR*) from *Fusobacterium nucleatum* W1481 were ordered in a pCW-lic backbone from GenScript (https://www.genscript.com/).

As controls, empty pCW-lic vectors were transformed into each knockout strain. The wild-type genes *gudP, garP, garL*, and *garR* from *E. coli* were also cloned into pCW-lic backbones under a *tac* promoter and transformed into their respective knockout strains.

**Chemical competency.** Keio collection of *E. coli* was made chemically competent using Mix & Go! Transformation Kit and Buffer (Zymo, T3001) according to the manufacturer's protocol and maintained at −80 °C until use.

**Transformation.** Each construct was individually transformed into chemically competent Keio collection *E. coli* according to the manufacturer's protocol (Zymo, T3001). The transformed cells were plated onto Luria-Bertani (LB) plates supplemented with 100 µg/mL carbenicillin to select for transformed colonies. Appropriate transformation was verified through Oxford Nanopore sequencing by Plasmidsaurus.

**Statistics and reproducibility.** No statistical method was used to predetermine sample size. No data were excluded from the analysis. The experiments were not randomized and investigators were not blinded to allocation during experiments and outcome assessment.

### Bioinformatic analysis

**Identification of gud/gar pathway in the GTDB genomes.** All representative genomes from the Genome Taxonomy Database (GTDB) (release r207) (Parks et al.[34]) were downloaded, and protein sequences for each genome were predicted using Prokka (version 1.14.6) (Seemann[47]). The HMM profiles for garD, garL, garR, gudD, and gudL were constructed. For garK, we used TIGR00045 retrieved from TIGRFAM[48]. Corresponding e-values were employed to search for homologs in the GTDB proteins using ProkFunFind. Details on the HMM profiles and e-values can be found at the provided GitHub link (https://github.com/frikinzi/sugar_oxidation_bioinfo).

The hits were filtered based on a maximum 1e-100 identity threshold, resulting in putative 3366 gud/gar sequences from 887 representative species. A genome was considered to have the minimum viable *gud/gar* pathway if it had GudD or GarD, GarR, and GudL or GarL.

**Structural prediction and molecular docking.** The structure for the putative *Enterocloster clostridioformis* GudL protein was predicted using AlphaFold2 (v2.3.0). Binding pockets were predicted using fpocket (v4.0) with default parameters. The pockets were compared to the homologous *Thermotoga maritima* dihydrodipicolinate synthase (3PB2) to identify putative substrate binding regions and catalytic residues[49]. The structure for 5-KDG (PubChem compound identifier: 5288442) was docked onto the predicted GudL structure using AutoDock Vina (v4.2)[50]. The docking simulation was performed within 20 Å × 20 Å × 20 Å cubes centered on the center points of the chosen fpocket substrate binding pocket with exhaustiveness set to 32. Docking results were visualized using PyMOL[51]. Catalytic residues were chosen based on previous studies on DapA. The predicted GudL protein structure was aligned with the *T. maritima* 3PB2 protein using TM-Align[52]. Protein sequence conservation of GudL was visualized using ConSurf based on the putative GudL clade (https://consurf.tau.ac.il/consurf_index.php).

**Profiling gud/gar pathway abundance in the human gut.** To characterize the abundance of gud/gar transcripts in IBD compared to non-IBD, Human metagenomic and metatranscriptomic reads from HMP2 and HPFS were downloaded from the NCBI SRA and mapped to a reference database constructed with these 3366 gud/gar genes using Bowtie2 v2.5.1. We filtered out samples with less than 5 million total reads. The number of reads mapped to the reference was normalized by the total number of reads per sample and then multiplied by 1 million to give counts per million (CPM). Patients with IBD were categorized based on whether they had Crohn's disease or ulcerative colitis. Using the Shapiro–Wilk normality test, we determined that both the MTX and MGX counts data did not follow a normal distribution (Shapiro–Wilk normality test: MGX $W = W = 0.40175$, p-value < 2.2e-16, MTX $W = 0.32301$, p-value < 2.2e-16). Therefore, we used a Mann–Whitney U test for each comparison among the 3 disease categories to see if counts of the *gud/gar* pathway genes significantly differed between the different groups. We did not correct the p-value and decided to not normalize based on species abundance because of the low number of reads mapped to the reference.

**Phylogenetic analyses.** Amino acid alignments for GudD/GarD/GarR/GarL/GudL found in the GTDB database were generated using ClustalW v2.1 using the BLASTp hits generated by ProkFunFind[53]. If the same genome had multiple hits for a gene, we selected the gene that had the lowest e-value to either the *E. clostridioformis* or *E. coli* gene. This resulted in 887 sequences in the GarR alignment, 809 sequences in the GarD alignment, 782 sequences in the GudD alignment, 507 GarL sequences, and 380 GudL sequences (Supplemental Data). The multiple sequence alignment was then trimmed using Goalign v0.3.7 by removing positions with over 97% gaps[54]. Gene trees for GudD, GarD, and GarR were created from these trimmed alignments using IQ-Tree

v2.2.0.3, with default parameters and 1000 bootstraps[55]. Ancestral sequence reconstruction was performed on the GudL tree using GRASP, with default parameters.

## Reporting summary

Further information on research design is available in the Nature Portfolio Reporting Summary linked to this article.

## Data availability

All genomic data analyzed in this study are available through the GTDB. Accession numbers can be found in the Supplementary Data file. All metagenomic datasets analyzed in the study are publicly available as part of the HMP2 and HPFS projects in Supplementary Data. All HMM profiles used in searching for gud/gar homologs are available in the GitHub repository: https://github.com/frikinzi/sugar_oxidation_bioinfo/tree/main/taxon_distribution_analyses/hmm_profiles (https://doi.org/10.5281/zenodo.14617111). Source data are provided with this paper.

## Code availability

All code used in the bioinformatic analyses is available here: https://github.com/frikinzi/sugar_oxidation_bioinfo.

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

## Acknowledgements
This study used the computational resources of the NIH HPC Biowulf cluster (http://hpc.nih.gov) and the UMIACS cluster at the University of Maryland's Center for Bioinformatics and Computational Biology (https://www.umiacs.umd.edu/). S.L. is supported by the NIH training grant T32-AI089621. A.K.J. and X.J. are supported by the Intramural Research Program of the NIH National Library of Medicine. B.H. is supported by R35-GM155208 and startup funding from the University of Maryland. The following reagents were obtained through BEI Resources, NIAID, and NIH as part of the Human Microbiome Project: *Clostridium citroniae* WAL-17108, HM-315; *Enterocloster clostridioformis* WAL-7855, HM-317. pCW-LIC was a gift from Cheryl Arrowsmith (Addgene plasmid # 26098; http://n2t.net/addgene:26098; RRID:Addgene_26098)

## Author contributions
S.L., B.H., and X.J. conceptualized the project. S.L., A.K.J., M.G., G.A., and G.M.N., performed the experiments and/or analyses. S.L. and A.K.J. wrote the paper.

## Competing interests
The authors declare no competing interests.
