## [Transparent Peer Review file · Nature Communications]

Convergent evolution of oxidized sugar metabolism in commensal and pathogenic microbes in the inflamed gut

Corresponding Author: Dr Brantley Hall

Version 0:

Reviewer comments:

Reviewer #1

(Remarks to the Author)
NCOMMS-24-34575-T

In this manuscript, Levy et al. present an interesting study demonstrating the prevalence of oxidized sugar utilization pathways in commensal bacteria, which were previously thought to primarily support pathogen colonization in inflammatory conditions. The authors identify genes essential for oxidized sugar utilization in the commensal *E. clostridioformis*, where GarABC and GudL can functionally replace GudP and GarL in pathogenic *E. coli*. Based on the enzymatically divergent yet functionally analogous nature of these genes, the authors suggest convergent evolution. Additionally, they associate these newly identified genes with inflammatory bowel disease (IBD). While this study broadens our understanding of these metabolic pathways, the conclusions rely on indirect evidence, and further experimental validation is needed to strengthen the story.

Major comments

1. While oxidized sugar levels increase post-antibiotic treatment and NOS2 expression is elevated in IBD, this does not conclusively imply that oxidized sugars increase in IBD patients. The authors should provide clearer evidence to support this link. Is there data or supporting evidence on NOS2 expression or oxidized sugar levels in IBD patients?
2. Please clarify how the authors selected IBD-associated species. Did this involve re-analyzing metagenomic sequencing datasets?
3. In Figure 2, the authors should test glucarate as a carbon source as well. Can galactose non-utilizers metabolize glucarate? Do these non-utilizers lack all genes involved in oxidized sugar metabolism, or only some?
4. In line 121-129, the rationale for identifying more genes in oxidized sugar metabolism should be clarified, particularly regarding GarK (which converts glycerol to 2-PG), as it is missing in *E. clostridioformis*. Is GarK not essential for using oxidized sugars as a carbon source?
5. While the bromothymol blue-based fermentation test is acceptable, it would be more robust if the authors conducted this test for the gudL and garABC complementation experiments, given that these genes are central to the study's findings on oxidized sugar fermentation.
6. Although the complementation experiment supports the functional role of GudL replacing GarL, the aldolase activity of GudL has not been directly shown. This may be challenging, as 5-KDG is not commercially available, but the authors could try using a K167A mutant to test if K167 is involved in catalysis.
- 6-1. Is the catalytic lysine conserved in GarL, as it is in GudL? This relates to the concern about GudL's aldolase activity.
7. The authors claim that *Fusobacteria* has adapted to inflammatory environments by acquiring the gene cluster for oxidized sugar metabolism. To strengthen this argument, the authors can provide experimental data demonstrating whether *Fusobacteria* can grow in the presence of oxidized sugars as the sole carbon source.
8. How does the abundance of the GarL gene in IBD patients compare to non-IBD individuals?

9. If the gud/gar pathway, previously considered a pathogen colonization factor, is widespread in commensals, it would be convincing to show that the expression of these genes is highly inducible. Are oxidized sugars or nitric oxide (NO) capable of inducing their expression?
10. If both *E. coli* with GarL and *E. clostridioformis* with GudL are present, which strain would have a competitive advantage? Alternatively, a Δ garL::gudL mutant outcompete Δ garL::garL, or vice versa?
11. The D-glucarate/galactarate utilization operon has been identified in *Bacillus subtilis* (Hosoya et al., FEMS Microbiology Letters, 2010). What is the abundance of these genes in IBD patients?

Minor comments

1. The title needs revision, as the authors have not experimentally shown adaptation to the gut.
2. The graphical abstract should be revised since no direct evidence of host inflammation effects has been provided.
3. The terms "parallel evolution" (title) and "convergent evolution" (main text) are used inconsistently. Please clarify the distinction.
4. In introduction line 60, the term should be "2-phosphoglycerate" (2-PG) rather than "2-phosphoglycerokinase."
5. Update bacterial taxa nomenclature, e.g., replace "Firmicutes" with "Bacillota (previously Firmicutes)." Check for consistency throughout the manuscript.
6. Ensure figures are presented in the same order they are mentioned in the text. Currently, Figure 2 is referenced before Figure 1.
7. The title of Figure 2 should be revised, as the authors did not appear to use minimal medium but instead used BHI medium with unidentified carbon sources. The phrase "galactarate as a sole carbon source" should be replaced with "galactarate as a carbon source."
8. In Figure 3A, Methods are described, but the results and interpretation for Figure 3A are missing in the text.
9. In line 166, please specify which panel of Figure 3 is being referred to.
10. Figures 4E and 4F are not described in the main text.
11. In line 194, define "RMSD" for general readers.
12. In line 228, the gene should be GarL, not Gud?
13. In lines 232-233, the authors mention 887 gut microbial species with the putative gud/gar pathway. It would be good to see whether these species are associated with IBD.
14. The authors suggest strain-specific functions for oxidized sugar metabolism (line 235). How does this apply to *E. coli*, given that some strains are commensal and others pathogenic?
15. In Figure 7A, the gudL gene in *E. coli* should be labeled as GarL?
16. In line 251, confirm whether it should be Fusobacteria or Fusobacteriota.
17. Ensure "sp." is not italicized.
18. Figure 8B is not cited in the main text.

Reviewer #2

(Remarks to the Author)

Levy et al. report on the taxonomic prevalence of proteins that facilitate oxidized sugar metabolism among gut commensal bacteria. The rationale for this investigation is that bacteria associated with inflammatory bowel diseases may gain a fitness advantage by metabolizing sugars that are spontaneously oxidized during inflammation. The authors additionally present functional and bioinformatic evidence for two protein components of oxidized sugar metabolism that differ from previously characterized systems, and that were subsequently found in many commensal bacterial species. The manuscript is well written and concise. Overall, the authors present a thorough account of Gud/Gar proteins among microbiota species and expand our understanding oxidized sugar metabolism.

Comments:

1. Figure 3. The images in Fig. 3A lack appropriate context. Is each panel from a well of a microtiter plate or something else?

If from a microtiter plate, color differences between strains should be compared across multiple wells in the same image or at least across an entire well. The text beginning on line 165 describes the Fig. 3 results as a measure of *E. coli* growth; however, there isn't a direct measurement of bacterial growth, so the interpretation should be limited to sugar fermentation here and in the Discussion.

2. Lines 181-183. Although the functional complementation data with GudL are convincing, the authors have not demonstrated a specific enzymatic activity for GudL. Therefore, claims about its catalytic activity should be presented as predictions in the absence of biochemical data.

3. Figure 4E and 4F do not appear to be discussed anywhere in the text.

4. The UC-CD comparison is not provided for Figure 8A. Are reads also significantly different between these two inflammatory conditions as they are for the gudL comparison? Does this result suggest differences in oxidized sugar metabolism between UC and CD or otherwise limit interpretation of the differences seen between inflamed and non-inflamed gut distribution?

5. Line 338. The statement that oxidized sugar metabolism was previously thought to only benefit pathogens is misleading since the pathway has been thoroughly characterized in commensal strains of *E. coli*.

6. The hypothesized fitness advantage provided to commensal inhabitants of the inflamed gut by oxidized sugar metabolism is challenged by the strain-to-strain variation of this pathway within species. Can the authors comment on this observation and whether alternative pathways may also exist?

7. The authors hypothesize that horizontal transfer contributes to the proliferation of gud/gar genes among different microbiota species. What is known about the prevalence of oxidized sugars beyond the context of the inflamed gut and potential origins of the two divergent systems identified here? Are there other environments in which galactarate and glucarate could be encountered and where catabolism of these sugars would be beneficial?

8. Related to the above point, are Gud/Gar proteins predicted for bacterial species that do not reside in the human gut. If so, what is the expected prevalence of this pathway outside the commensal microbiota?

Minor comments:

1. The Annotation key for Fig. 1B is redundant since each of the genes are labeled in the ORF map.

2. The panel labels for Figure 8 do not correspond to those used in the text. See 8B and 8C, lines 279 and 284.

Version 1:

Reviewer comments:

Reviewer #1

(Remarks to the Author)

I appreciate the authors for their extensive revisions, which have significantly strengthened their findings.

I have only one minor comment related to my previous comment #7. For Figures 3, 5, and 7, I believe the phrase 'a sole carbon source' would be better replaced with 'a carbon source.'

Reviewer #2

(Remarks to the Author)

The revised manuscript by Levy et al. addresses the critiques raised in my original assessment and I have no additional comments.

Response to Referees

Responses to each of the reviewer's comments/suggestions are provided in blue text below each point and quotes from the text are included as purple text. Lines numbers are given for the revision locations within the revised text.

Reviewer #1 (Remarks to the Author):

NCOMMS-24-34575-T

In this manuscript, Levy et al. present an interesting study demonstrating the prevalence of oxidized sugar utilization pathways in commensal bacteria, which were previously thought to primarily support pathogen colonization in inflammatory conditions. The authors identify genes essential for oxidized sugar utilization in the commensal *E. clostridioformis*, where GarABC and GudL can functionally replace GudP and GarL in pathogenic *E. coli*. Based on the enzymatically divergent yet functionally analogous nature of these genes, the authors suggest convergent evolution. Additionally, they associate these newly identified genes with inflammatory bowel disease (IBD). While this study broadens our understanding of these metabolic pathways, the conclusions rely on indirect evidence, and further experimental validation is needed to strengthen the story.

Major comments

1. While oxidized sugar levels increase post-antibiotic treatment and *NOS2* expression is elevated in IBD, this does not conclusively imply that oxidized sugars increase in IBD patients. The authors should provide clearer evidence to support this link. Is there data or supporting evidence on *NOS2* expression or oxidized sugar levels in IBD patients?

We appreciate the opportunity to provide additional evidence to support the premise of our manuscript. In HMP2, researchers compared gene expression levels in the ileum and rectum of IBD patients. They found that the *NOS2* expression log fold change was 2.11-2.42 in CD patients and 3.58 in UC patients compared to non-IBD controls (Supplementary Table 31, The Integrative HMP (iHMP) Research Network Consortium 2019, <https://doi.org/10.1038/s41586-019-1238-8>). Other genes that contribute to the high levels of reactive oxygen and nitrogen species in the gut are also upregulated during IBD, including *DUOX2* (logFC ~ 3.5). Furthermore, the PRISM cohort stool metabolomics found that both glucarate and galactarate were increased in abundance in IBD patients (Franzosa 2019, <https://doi.org/10.1038/s41564-018-0306-4>). We believe that these comprehensive

transcriptomics and metabolomics studies sufficiently demonstrate the increased *NOS2* expression and concordant oxidized sugar abundance in IBD patients.

We have added the following text (Lines 49-55)

“The Integrative Human Microbiome Project reported 4-12 fold higher *NOS2* expression in those with active flares of IBD compared to healthy individuals.¹⁶⁻²⁰ In colonic epithelial cells, *NOS2* expression produces nitric oxide radicals that diffuse directly into the lumen and spontaneously oxidize glucose and galactose to glucarate (saccharic acid) and galactarate (mucic acid), respectively. PRISM metabolomics revealed concordantly high abundance oxidized sugars in the inflamed IBD gut.^{21,22}”

2. Please clarify how the authors selected IBD-associated species. Did this involve re-analyzing metagenomic sequencing datasets?

We thank the reviewer for highlighting this point, which does require clarification. When selecting IBD-associated species to screen for oxidized sugar metabolism, we began by referring to Schirmer et al. 2018, “Microbial genes and pathways in inflammatory bowel disease.” In this manuscript, Schirmer et al. performed a meta-analysis of bacterial species identified in PRISM, LSS, Lewis, and NLIBD cohorts, and identified bacterial species found to be increased in IBD patients within each cohort, and across studies. We chose species that both increased during IBD in at least one study and lacked a known mechanism behind their increase in relative abundance during inflammation.

We have added the following text (Lines 107-109)

“To investigate this theory, we screened a selection of species that were both increased in relative abundance during IBD, and lacked a known mechanism behind their increased abundance for growth on oxidized sugars as a carbon source.²⁹”

3. In Figure 2, the authors should test glucarate as a carbon source as well. Can galactose non-utilizers metabolize glucarate? Do these non-utilizers lack all genes involved in oxidized sugar metabolism, or only some?

We appreciate this insight from the reviewer, and have tested both galactose and glucarate as a carbon source, resulting in a more comprehensive screen that evaluates growth with glucose, galactose, glucarate, or galactarate as an added carbon source (Figure 2A-F). Of the species we screened for oxidized sugar metabolism, all of them contained the genes necessary for galactose utilization (*galE*, *galT*, *galK*, and *galM*) and demonstrated enhanced growth on galactose as a carbon source. The three species

that showed growth benefits from galactarate as a carbon source also showed similar benefits from glucarate as a carbon source in the new experiment.

The species that were unable to utilize oxidized sugars lacked most of the genes involved in oxidized sugar metabolism, but some did contain homologs. For example, in the *C. symbiosum* genome, we searched against the curated HMM profile for each *gud/gar* gene, and identified homologs to GarL (e-value $4e-73$) and GarR (e-value $2.7e-58$). Additionally, *M. lactaris* and *B. fragilis* contain homologs to GudL (e-value $2.2e-48$ and $5.8e-47$, respectively). However, none of these homologs met our threshold of $1e-100$.

4. In line 121-129, the rationale for identifying more genes in oxidized sugar metabolism should be clarified, particularly regarding GarK (which converts glycerol to 2-PG), as it is missing in *E. clostridioformis*. Is GarK not essential for using oxidized sugars as a carbon source?

This is an excellent suggestion from the reviewer, as we did not provide sufficient insight into our rationale behind the genes selected for the minimum viable pathway necessary for oxidized sugar metabolism. When searching for *garK*, we identified homologs in *E. clostridioformis* and *M. lactaris*. However, *E. clostridioformis* contained 2 copies of *garK* homologs, both located outside of the *gud/gar* gene cluster that contained *gudD*, *garD*, *garR*, the putative *gudL* and *garABC*. Furthermore, the gene tree for *garK* did not follow a similar topology to the gene trees for the other *gud/gar* genes that were deemed part of the minimum viable pathway (*gudD/garD*, *gudL/garL*, and *garR*)

(Supplementary Figure 3). Upon constructing the *garK* gene tree, we found that *garK* was polyphyletic, suggesting independent recruitment of kinases multiple times throughout evolution. This pattern contrasts sharply with the other *gud/gar* genes (*gudD/garD*, *gudL/garL*, and *garR*), which show consistent co-evolution as a single functional unit.

When looking for the *gud/gar* pathway in specific strains, we noticed that many strains contained *garK* homologs without containing any of the other *gud/gar* genes, and other strains would contain *gud/gar* without *garK* (Supplementary Figure 3).

Kinases are known to be promiscuous and multifunctional. Other glycerate kinases can metabolize glycerate into products other than 2-PG, like glycerate-3-kinase from *E. coli*, which catalyzes the reaction of glycerate to 3-PG (Doughty 1996, PMID: 5325263).

Therefore, *garK* was not selected as a key gene in this study.

We have added the following text to clarify our thought process and elucidate our rationale: (Lines 124-128)

“Notably, BLASTp only revealed proteins in *E. clostridioformis* homologous to GudD (percent identity 64.3%, e-value 7.37e-221, bitscore 611), GarD (percent identity 53.9%, e-value 5.11e-191, bitscore 541), GarR (percent identity 67.8%, e-value 2.27e-140, bitscore 395), and two copies of GarK (percent identity 43.5%, e-value 2.14e-98, bitscore 285; percent identity 42.3%, e-value 1.22e-100, bitscore 291).”

And (Lines 246-251)

“GarK was excluded from the minimum viable pathway, as upon constructing the *garK* gene tree, we found that *garK* was polyphyletic, suggesting independent recruitment of kinases multiple times throughout evolution. This pattern contrasts sharply with the other *gud/gar* genes (*gudD/garD*, *gudL/garL*, and *garR*), which show consistent co-evolution as a single functional unit (Supplementary Figure 3).”

5. While the bromothymol blue-based fermentation test is acceptable, it would be more robust if the authors conducted this test for the *gudL* and *garABC* complementation experiments, given that these genes are central to the study’s findings on oxidized sugar fermentation.

We thank the reviewer for their feedback. We have updated Figure 3A to better reflect the findings regarding GudL by showing the color change from *garL* and *gudL* mutants and complements, and conducted the fermentation test for *gudL* complements on galactarate (Figure 3C) and glucarate (Figure 3F). We have also conducted the bromothymol-blue based fermentation test on Δ *gudP::garABC* and Δ *garP::garABC* complements (Figure 3B, E).

6. Although the complementation experiment supports the functional role of GudL replacing GarL, the aldolase activity of GudL has not been directly shown. This may be challenging, as 5-KDG is not commercially available, but the authors could try using a K167A mutant to test if K167 is involved in catalysis.

We thank the reviewer for their suggestions on how to strengthen our argument regarding GudL activity. The reviewer is correct that we are unable to acquire 5-KDG.

Therefore, we performed the K167A mutation suggested, as well as others, to demonstrate the catalytic activity of key residues. The K167A mutation abolished metabolism of both glucarate and galactarate (Figure 5D, E). Mutations of key residues to the ancestral states K194N and M110Y also abolished oxidized sugar metabolism, while mutation to a neutral residue K194A merely reduced oxidized sugar fermentation (Figure 5D, E). Though this does not directly demonstrate the aldolase activity of GudL, we believe that these experiments validate the key residues and demonstrate that a mutation to the ancestral DapA residue does abolish GudL activity.

We have added the following text (Lines 216-225):

“Notably, there was a predicted change from Y110 to M110 and N194 to K194 from the ancestral state to the GudL clade, which are universally conserved among the putative GudL clade. These residues unique to the GudL clade may contribute to its function, and delineate it from other DapA-like enzymes. We mutated the *E. clostridioformis* GudL sequence to change position M110 and K194 to a neutral amino acid with no side chain (M110A, K194A), and to their ancestral states (M110Y, K194N). Mutating these residues to alanine abolished fermentation of both glucarate and galactarate, with the exception of K194A, which reduced but did not abolish oxidized sugar metabolism (Figure 5D and E). These results indicate that the GudL clade can be delineated from other DapA at node 1.”

and (Lines 231-234):

“To verify the importance of K167, we mutated the GudL sequence to change position K167 into alanine. The mutation abolished utilization of both glucarate and galactarate (Figure 5D and E).”

6-1. Is the catalytic lysine conserved in GarL, as it is in GudL? This relates to the concern about GudL's aldolase activity.

The catalytic residues in GarL are not conserved between GudL and GarL. In GarL, the key catalytic residues are His50, Arg75, Asp89, Glu153, and Asp179, while in GudL, the key residue is K167. A structural alignment demonstrates that these key residues do not align. However, we were able to demonstrate that mutation of K167 to A167 abolished growth on oxidized sugars (Figure 5D, E)

7. The authors claim that *Fusobacteria* has adapted to inflammatory environments by acquiring the gene cluster for oxidized sugar metabolism. To strengthen this argument, the authors can provide experimental data demonstrating whether *Fusobacteria* can grow in the presence of oxidized sugars as the sole carbon source.

We thank the reviewer for this suggestion. Unfortunately, the strains of *Fusobacteria* that contain the *gud/gar* pathway are unavailable from BEI or ATCC, and so we were unable to validate that *Fusobacteria* can grow on oxidized sugars as a carbon source. To compensate, we heterologously expressed a *Fusobacteria* *gudL* and *garR* in *E. coli* *garL* and *garR* knockouts, respectively, and showed that complementation with the *Fusobacteria* genes partially restored oxidized sugar fermentation on both glucarate and galactarate as a sole carbon source (Figure 7C, D).

We have added the following text (Lines 282-284)

“To verify the functionality of the *Fusobacterium nucleatum* *gud/gar* genes, we complemented Keio *E. coli* knockouts with the *F. nucleatum* genes *gudL* and *garR*, which partially restored fermentation of both glucarate and galactarate (Figure 7C, D).”

8. How does the abundance of the *GarL* gene in IBD patients compare to non-IBD individuals?

This is an intriguing inquiry by the reviewer. We performed metagenomic and metatranscriptomic analysis to examine the relative abundance and expression of *garL* in IBD compared to non-IBD individuals (Supplementary Figure 4). We found that *garL* is increased in relative abundance in Crohn’s Disease (CD) patients compared to non-IBD controls, and identified increased relative abundance of *garL* transcripts. However, *garL* was not significantly different in Ulcerative Colitis (UC) patients compared to healthy controls.

We have added the following text (Lines 313-318)

“When we examined changes in *garL* between IBD patients and non-IBD individuals, we found that *garL* genes and transcripts were increased in CD patients, but not in UC patients compared to healthy controls (Supplementary Figure 5) (Mann Whitney U Test, MGX: CD $p=8.808e-05$ $W=115671$, UC $p=.07631$ $W=56227$, UC-CD $p=.03518$ $W=102641$, MTX: CD $p=1.752e-08$ $W=99977$, UC $p=0.2358$ $W=57046$, UC-CD $p=0.005956$ $W=28776$).”

9. If the *gud/gar* pathway, previously considered a pathogen colonization factor, is widespread in commensals, it would be convincing to show that the expression of these

genes is highly inducible. Are oxidized sugars or nitric oxide (NO) capable of inducing their expression?

We thank the reviewer for their insightful question. It has been previously shown that in *E. coli*, GudD, GarD, GarR, and GarK activity can be induced by galactarate and glucarate, but not by other substrates like glycerol (Monterrubbio, 2000, <https://doi.org/10.1128/jb.182.9.2672-2674.2000>). The inducibility of the novel *gud/gar* enzymes in gram-positive bacteria is an excellent avenue for future work.

10. If both *E. coli* with GarL and *E. clostridioformis* with GudL are present, which strain would have a competitive advantage? Alternatively, a Δ garL::gudL mutant outcompete Δ garL::garL, or vice versa?

We appreciate this thoughtful inquiry from the reviewer. Determining whether *E. coli* with GarL or *E. clostridioformis* with GudL would have a competitive advantage in the gut would be quite challenging, due to the number of other factors present in the environment. In the gut, anaerobic species like *E. clostridioformis* usually have an advantage, but during gut inflammation, availability of alternative electron acceptors like nitrate, nitrite, and even oxygen favors species like *E. coli* that can exploit the oxidizing environment. While our study does not evaluate the relative efficiency of GudL compared to GarL, the impact of these enzymes on the growth and competitive dynamics of their hosts in an inflamed gut environment presents an intriguing avenue for future research.

11. The D-glucarate/galactarate utilization operon has been identified in *Bacillus subtilis* (Hosoya et al., FEMS Microbiology Letters, 2010). What is the abundance of these genes in IBD patients?

We appreciate the reviewer's valuable perspective on the *gud/gar* pathway found in *B. subtilis*. This pathway is partially homologous to the pathway in *E. coli*, using YcbF and YcbH (homologous to GudD and GarD) to form 5-keto-4-deoxyglucarate, similar to the *E. coli gud/gar* pathway. From there, the pathway diverges, as *B. subtilis* may use Ycb to produce 2,5-dioxopentanoate.

When performing metagenomic and metatranscriptomic analyses on the *B. subtilis ycb* pathway, we built a reference database using *B. subtilis ycb* homologs, yielding 1,112 sequences. While we found increased abundance of *ycb* in CD patients compared to non-IBD controls, very few reads mapped to the *ycb* pathway for UC and non-IBD samples (Mann-Whitney U Test, UC vs nonIBD, $W = 55318$, $p\text{-value} = 0.464$; CD vs nonIBD, $W = 92523$, $p\text{-value} = 0.05134$; CD vs UC, $W = 92523$, $p\text{-value} = 0.05134$). Of the 148 bacterial species predicted by ProkFunFind to contain the *ycb*

pathway, only 12 are annotated to inhabit the animal gut, supporting the low read mapping to *ycb* genes in the gut environment.

This indicates that the *B. subtilis ycb* pathway is not the primary pathway present in the gut environment. Since *B. subtilis* is a soil microbe, we would not expect it to be found in the gut environment, and the results from our metagenomic analysis supports the rarity of this alternative *ycb* pathway in the gut.

Minor comments

1. The title needs revision, as the authors have not experimentally shown adaptation to the gut.

We have revised the title of the manuscript to “Convergent evolution of oxidized sugar metabolism in commensal and pathogenic microbes in the inflamed gut.”

2. The graphical abstract should be revised since no direct evidence of host inflammation effects has been provided.

We appreciate the reviewer’s input and agree that we did not demonstrate direct evidence of host inflammation effects. We have removed the graphical abstract from the manuscript.

3. The terms "parallel evolution" (title) and "convergent evolution" (main text) are used inconsistently. Please clarify the distinction.

We agree with the reviewer that their usage was inconsistent and confusing. All instances of “parallel evolution” have been revised to “convergent evolution” for clarity.

4. In introduction line 60, the term should be "2-phosphoglycerate" (2-PG) rather than "2-phosphoglycerokinase."

We appreciate the reviewer’s attention to detail. The line has been corrected.

5. Update bacterial taxa nomenclature, e.g., replace "Firmicutes" with "Bacillota (previously Firmicutes)." Check for consistency throughout the manuscript. All instances of bacterial nomenclature have been updated to the current version, with references to the previous nomenclature at the first instance.

6. Ensure figures are presented in the same order they are mentioned in the text. Currently, Figure 2 is referenced before Figure 1.

We have revised the text to ensure the figures are presented and referenced in the same order. Now, Figure 1 is referenced first in lines 67-73.

“The well-characterized *E. coli* *gud/gar* pathway contains five core genes: glucarate/galactarate permeases (*gudP/garP*) imports the oxidized sugars, glucarate/galactarate dehydratases (*gudD/garD*) convert the oxidized sugars into a common 5-keto-4-deoxy-D-glucarate (5-KDG) intermediate, 5-KDG aldolase (*garL*) splits 5-KDG into pyruvate and tartronate semialdehyde (TSA), TSA dehydrogenase (*garR*) forms glycerate, and, as a final step, glycerate kinase (*garK*) produces 2-PG (Figure 1A).^{23,27,28}”

7. The title of Figure 2 should be revised, as the authors did not appear to use minimal medium but instead used BHI medium with unidentified carbon sources. The phrase “galactarate as a sole carbon source” should be replaced with “galactarate as a carbon source.”

This is an excellent point from the reviewer. The title of Figure 2 has been revised as suggested, and now reads “Identification of gut microbial species increased during inflammation capable of metabolizing glucarate or galactarate as a carbon source” to more accurately reflect the experimental design.

8. In Figure 3A, Methods are described, but the results and interpretation for Figure 3A are missing in the text.

We have added references to Figure 3A in the text to provide interpretation of the results.

We have revised the following text (Lines 169-174)

“Knockouts of *E. coli* *gudP* and *garL* abolished fermentation of glucarate and galactarate carbon sources, while deletion of *garP* and *garR* attenuated oxidized sugar fermentation (Figure 3A-G, Supplementary Figure 2). Complementation of the native *E. coli* gene back into the mutant strain restored utilization of glucarate and galactarate to that of the wild-type *E. coli* in all cases, validating the ectopic expression methodology (Figure 3A-G, Supplementary Figure 3).”

and (Lines 179-181)

“Most notably, cross-species complementation of the *E. coli* Δ *garL* mutants with the putative *E. clostridioformis* aldolase *gudL* restored fermentation of glucarate and galactarate as strongly as isogenic complementation (Figure 3A, C, F).”

9. In line 166, please specify which panel of Figure 3 is being referred to.

The text has been revised to refer to Figure 3A-G.

10. Figures 4E and 4F are not described in the main text.

We have referenced Figures 4E and 4F in the text (Lines 255-257):

“Interestingly, *GarL* was primarily found in Pseudomonadota, while *GudL* is widely spread among a diverse range of phyla, including Bacillota and Fusobacteriota (Figure 4E, F).”

11. In line 194, define "RMSD" for general readers.

RMSD has been defined as root mean square deviation.

12. In line 228, the gene should be *GarL*, not *Gud*?

We appreciate the reviewer catching this. The enzyme name has been corrected to *GarL* instead of *GudL*.

13. In lines 232-233, the authors mention 887 gut microbial species with the putative *gud/gar* pathway. It would be good to see whether these species are associated with IBD.

We thank the reviewer for bringing this to our attention, as this would be an interesting association to uncover. In our work, we discovered that the *gud/gar* pathway is highly strain specific. This means that while some species we identified as containing the *gud/gar* pathway - *Enterocloster bolteae*, *Enterocloster clostridioformis*, *Fusobacterium nucleatum*, *Fusobacterium mortiferum*, *Blautia producta*, *Clostridium symbiosum*, and *Escherichia coli* to list a few - have been found to be associated with IBD, there is limited information on the strain-level microbial associations with IBD. This knowledge gap makes it difficult to translate the prevalence of our strain-specific pathway to the broader species-level information available regarding bacterial IBD associations.

14. The authors suggest strain-specific functions for oxidized sugar metabolism (line 235). How does this apply to *E. coli*, given that some strains are commensal and others pathogenic?

The implication of oxidized sugar metabolism in some *E. coli* strains is an interesting question from the reviewer. We hypothesize that both commensal and pathogenic strains would benefit from the ability to metabolize oxidized sugars produced in the inflamed gut. During IBD, *E. coli* blooms are common (Khorsand 2022, <https://doi.org/10.3389/fcimb.2022.1015890>; Kittana 2023, <https://doi.org/10.1128/msphere.00478-22>). However, it is difficult to discern whether the blooms are caused by the increased abundance of nitrate to act as a terminal electron acceptor, the increased abundance of oxidized sugars used as an alternative carbon source, or other factors.

To look at strain specificity, we examined a published pangenome of *Escherichia coli* (Tantoso 2022, <https://doi.org/10.1186/s12915-022-01347-7>), which contains both pathogenic and commensal strains, and found that the *gud/gar* genes are functionally ubiquitous in *E. coli* genomes (*gudD*: 1321/1324 genomes, *garD*: 1295/1324 genomes, *garL*: 1324/1324 genomes, *garR*: 1323/1324 genomes, *garK*: 1323/1324 genomes), suggesting that, while strain-specific in some species, it is a core function of both commensal and pathogenic *E. coli*.

15. In Figure 7A, the *gudL* gene in *E. coli* should be labeled as GarL?

We appreciate the reviewer's correction. The legend has been revised to refer to GarL and GudL instead of GudL and GudL.

16. In line 251, confirm whether it should be Fusobacteria or Fusobacteriota.

All instances referring to the Fusobacteriota phylum as Fusobacteria have been updated to the appropriate "Fusobacteriota" nomenclature.

17. Ensure "sp." is not italicized.

We have confirmed that all instances of sp. are no longer italicized.

18. Figure 8B is not cited in the main text.

Figure 8B is now cited in Line 308-313.

"To do this, we used 380 putative *gudL* homologs as a reference database, and determined that *gudL* was increased in patients with IBD for both the metatranscriptomes and metagenomes (Figure 8B, 8D) (Mann Whitney U Test, MGX: CD $p=3.331e-10$ $W=126260$, UC $p=0.0002354$ $W=50742$, UC-CD $p=0.008988$ $W=105092$, MTX: CD $p<2.2e-16$ $W=123662$, UC $p=1.614e-14$ $W=38031$, UC-CD $p=0.7831$ $W=27261$).

Reviewer #2 (Remarks to the Author):

Levy et al. report on the taxonomic prevalence of proteins that facilitate oxidized sugar metabolism among gut commensal bacteria. The rationale for this investigation is that bacteria associated with inflammatory bowel diseases may gain a fitness advantage by metabolizing sugars that are spontaneously oxidized during inflammation. The authors additionally present functional and bioinformatic evidence for two protein components of oxidized sugar metabolism that differ from previously characterized systems, and that were subsequently found in many commensal bacterial species. The manuscript is well written and concise. Overall, the authors present a thorough account of Gud/Gar proteins among microbiota species and expand our understanding of oxidized sugar metabolism.

Comments:

1. Figure 3. The images in Fig. 3A lack appropriate context. Is each panel from a well of a microtiter plate or something else? If from a microtiter plate, color differences between strains should be compared across multiple wells in the same image or at least across an entire well. The text beginning on line 165 describes the Fig. 3 results as a measure of *E. coli* growth; however, there isn't a direct measurement of bacterial growth, so the interpretation should be limited to sugar fermentation here and in the Discussion.

We thank the reviewer for their feedback, and agree that additional context is necessary. We have updated Figure 3A to provide a better view of the 96-well plate, and provided the entirety of the before and after images in Supplemental Figure 2 to help give appropriate context. Additionally, we have reframed references to bacterial growth to more accurately describe color change as oxidized sugar utilization, metabolism and fermentation.

2. Lines 181-183. Although the functional complementation data with GudL are convincing, the authors have not demonstrated a specific enzymatic activity for GudL. Therefore, claims about its catalytic activity should be presented as predictions in the absence of biochemical data.

We appreciate the insight, and agree that with the data provided in the initial submission, claims about catalytic activity should have been predictions. To support our claims about catalytic activity, we performed mutations of key predicted catalytic

residues. By comparing GudL to the ancestral DapA protein, we identified putative key residues K167, M110, and K194. We mutated these residues to neutral residue alanine and back to the ancestral residue (K167A, M110A, M110Y, K194A, K194N). Of these, all but K194A abolished oxidized sugar fermentation (Figure 5D, E). With this data, we are confident that GudL catalyzes the same step in the oxidized sugar metabolic pathway as GarL, although due to the unavailability of 5-KDG, we were not able to directly demonstrate its aldolase functionality.

We have added the following text (Lines 218-225)

“These residues unique to the GudL clade may contribute to its function, and delineate it from other DapA-like enzymes. We mutated the *E. clostridioformis* GudL sequence to change position M110 and K194 to a neutral amino acid with no side chain (M110A, K194A), and to their ancestral states (M110Y, K194N). Mutating these residues to alanine abolished fermentation of both glucarate and galactarate, with the exception of K194A, which reduced but did not abolish oxidized sugar metabolism (Figure 5D and E). These results indicate that the GudL clade can be delineated from other DapA at node 1.”

3. Figure 4E and 4F do not appear to be discussed anywhere in the text.

We appreciate the reviewer’s attention to detail, and have added a reference to Figures 4E and F.

We have added the following text (Lines X-Y)

“Interestingly, GarL was primarily found in Pseudomonadota, while GudL is widely spread among a diverse range of phyla, including Bacillota and Fusobacteriota (Figure 4E, F).”

4. The UC-CD comparison is not provided for Figure 8A. Are reads also significantly different between these two inflammatory conditions as they are for the gudL comparison? Does this result suggest differences in oxidized sugar metabolism between UC and CD or otherwise limit interpretation of the differences seen between inflamed and non-inflamed gut distribution?

We thank the reviewer for noting this oversight. We have added the UC-CD comparison in Line 298 ($p=0.008988$ $W=105092$). Interestingly, for both the *gud/gar* pathway as a whole and *gudL*, a significant difference between UC and CD patients is seen in metagenomic data, but not metatranscriptomics.

5. Line 338. The statement that oxidized sugar metabolism was previously thought to only benefit pathogens is misleading since the pathway has been thoroughly characterized in commensal strains of *E. coli*.

We appreciate the reviewer's attention to detail. We have removed the misleading statement.

6. The hypothesized fitness advantage provided to commensal inhabitants of the inflamed gut by oxidized sugar metabolism is challenged by the strain-to-strain variation of this pathway within species. Can the authors comment on this observation and whether alternative pathways may also exist?

The reviewer raises an excellent point. Alternative oxidized sugar metabolic pathways do exist, as exemplified by the *ycb* pathway identified in soil microbe *Bacillus subtilis* (Hosoya 2002, [https://doi.org/10.1016/S0378-1097\(02\)00612-2](https://doi.org/10.1016/S0378-1097(02)00612-2)). This pathway is partially homologous to the pathway in *E. coli*, using YcbF and YcbH (homologous to GudD and GarD) to form 5-keto-4-deoxyglucarate, similar to the *E. coli gud/gar* pathway. From there, the pathway diverges, as *B. subtilis* may use Ycb to produce 2,5-dioxopentanoate. The existence of at least three varying metabolic pathways for oxidized sugar utilization (*E. coli gud/gar*, *E. clostridioformis gud/gar*, and *B. subtilis ycb*) suggests that additional pathways may exist.

Though we do suggest that the ability to metabolize oxidized sugars as an alternative carbon source may provide a competitive advantage in the inflamed gut, there are numerous other factors that contribute to changes in microbial abundance. The strain-specificity of *gud/gar* implies that the competitive advantage conferred may not outweigh other contributing factors, like aerobic resilience. Strain-specific features can arise from instances of recent horizontal gene transfer, like the putative instance of Bacillota to Fusobacteriota, in which GarR has a shared percent identity of 82% between *Enterococcus D. casseliflavus* and *Fusobacterium B. sp.*.

7. The authors hypothesize that horizontal transfer contributes to the proliferation of *gud/gar* genes among different microbiota species. What is known about the prevalence of oxidized sugars beyond the context of the inflamed gut and potential origins of the two divergent systems identified here? Are there other environments in which galactarate and glucarate could be encountered and where catabolism of these sugars would be beneficial?

We thank the reviewer for their thought-provoking question. Oxidized sugars, particularly glucarate, can be found in fruits and some cruciferous vegetables (Walaszek 1996, [https://doi.org/10.1016/0271-5317\(96\)00045-0](https://doi.org/10.1016/0271-5317(96)00045-0)). Due to the natural cycle of plant

growth and degradation, oxidized sugars can also be found in the soil. Therefore in both of these environments, the ability to metabolize oxidized sugars could be beneficial. However, one of the primary benefits to oxidized sugar metabolism in the gut - utilization as an alternative carbon source - is not present in these environments. In the gut, simple sugars like glucose and galactose are in low supply and is one of the limited factors to bacterial growth, which is why we hypothesize that the ability to utilize oxidized sugars as an alternative carbon source is so beneficial (Ferraris 1990, <https://doi.org/10.1152/ajpgi.1990.259.5.G822>; Gromova 2021, <https://doi.org/10.3390/nu13072474>). However, in high-sugar environments like fruit, there is less competition for alternative carbon sources, as simple sugars are plentiful. The concentration of simple sugars in soil ranges widely, and so it is difficult to speak to the environmental pressures that may have led to the development of oxidized sugar metabolism outside the gut (Reishke 2015, <https://doi.org/10.1016/j.soilbio.2014.10.012>).

8. Related to the above point, are Gud/Gar proteins predicted for bacterial species that do not reside in the human gut. If so, what is the expected prevalence of this pathway outside the commensal microbiota?

We thank the reviewer for the comment. We annotated the habitat of the species predicted to contain the *gud/gar* pathway using information from NCBI. Of the 887, 351 of our species reside in the gut. The other 536 have a variety of habitats, including terrestrial and wastewater environments. Other mechanisms for oxidized sugar metabolism, like the aforementioned *B. subtilis ycb* pathway, are prevalent in soil habitats.

Minor comments:

1. The Annotation key for Fig. 1B is redundant since each of the genes are labeled in the ORF map.

We appreciate the reviewer's attention to detail and have removed the redundant annotation key.

2. The panel labels for Figure 8 do not correspond to those used in the text. See 8B and 8C, lines 279 and 284.

We thank the reviewer for pointing this out and have corrected the in-text references.